# Towards Implicit Aggregation: Robust Image Representation for Place Recognition in the Transformer Era

**Feng Lu**[1,2*] **Tong Jin**[3,4*] **Canming Ye**[1] **Yunpeng Liu**[3†] **Xiangyuan Lan**[2,5†] **Chun Yuan**[1†]
[1]Tsinghua Shenzhen International Graduate School, Tsinghua University
[2]Pengcheng Laboratory [3]Shenyang Institute of Automation, Chinese Academy of Sciences
[4]University of Chinese Academy of Sciences [5]Pazhou Laboratory (Huangpu)
lufengrv@gmail.com jintong@sia.cn ycm24@mails.tsinghua.edu.cn
ypliu@sia.cn lanxy@pcl.ac.cn yuanc@sz.tsinghua.edu.cn

## Abstract

Visual place recognition (VPR) is typically regarded as a specific image retrieval task, whose core lies in representing images as global descriptors. Over the past decade, dominant VPR methods (*e.g.*, NetVLAD) have followed a paradigm that first extracts the patch features/tokens of the input image using a backbone, and then aggregates these patch features into a global descriptor via an aggregator. This backbone-plus-aggregator paradigm has achieved overwhelming dominance in the CNN era and remains widely used in transformer-based models. In this paper, however, we argue that a dedicated aggregator is not necessary in the transformer era, that is, we can obtain robust global descriptors only with the backbone. Specifically, we introduce some learnable aggregation tokens, which are prepended to the patch tokens before a particular transformer block. All these tokens will be jointly processed and interact globally via the intrinsic self-attention mechanism, implicitly aggregating useful information within the patch tokens to the aggregation tokens. Finally, we only take these aggregation tokens from the last output tokens and concatenate them as the global representation. Although implicit aggregation can provide robust global descriptors in an extremely simple manner, where and how to insert additional tokens, as well as the initialization of tokens, remains an open issue worthy of further exploration. To this end, we also propose the optimal token insertion strategy and token initialization method derived from empirical studies. Experimental results show that our method outperforms state-of-the-art methods on several VPR datasets with higher efficiency and ranks 1st on the MSLS challenge leaderboard. The code is available at `https://github.com/lu-feng/image`.

## 1 Introduction

Visual place recognition (VPR) involves identifying the coarse geographical location of a query place image by retrieving the most similar images from a geo-tagged database captured at previously visited places [46]. It is a fundamental and essential task in a wide range of computer vision and robotics applications, *e.g.*, augmented reality [53], autonomous driving [21], and SLAM [15]. Thus, it has garnered significant attention and study. Despite recent advances, there still exist some challenges in VPR, including condition variations, viewpoint changes, and perceptual aliasing (images from different places showing high similarity) [46], etc.

---

*Equal contribution.
†Corresponding authors.

39th Conference on Neural Information Processing Systems (NeurIPS 2025).

Typically, VPR is formulated as an image retrieval problem. For a given query image and a database, all place images are represented using global features, and the nearest neighbor search is conducted in this feature space to get the target place images that best match the query. The global features are usually obtained by employing aggregation methods (*e.g.*, VLAD [35]) to process local features. With the advancement of deep learning, most VPR methods have used a convolutional neural network (CNN) [31] or vision transformer (ViT) [23] as the backbone to extract local (patch) features. Meanwhile, NetVLAD [4] and GeM pooling [56] have become the most popular aggregation methods for aggregating local features to yield global descriptors, which are generally robust against common visual variations. Following this paradigm, some recent studies proposed more aggregation methods (*e.g.*, MixVPR [2], SALAD [34], CricaVPR [49], BoQ [3], and EDTformer [37]), trying to make the global features condition- and viewpoint-invariant, thereby achieving a promising performance.

Although this backbone-plus-aggregator VPR paradigm to obtain global features has become the de-facto standard [10] in the CNN era, it has some potential issues. First, the two-stage process (feature extraction + aggregation) may lead to unnecessary structural complexity and redundancy. Second, the one-shot aggregation of patch features implemented by the aggregator offers no opportunity for correction and refinement. Regarding specific aggregation methods (aggregators), there may exist some particular issues, such as the loss of position information of original patch features in NetVLAD [4]. Designing a perfect aggregator artificially is highly challenging. However, in light of the nature of transformer-based backbones, which are capable of modeling global contextual information and long-range dependencies [24], we argue that it is no longer necessary to design an aggregator separately. Instead, we can leverage the intrinsic self-attention mechanism within the backbone to implicitly aggregate useful information from patch tokens, thereby eliminating the need for an extra aggregator. In fact, previous work BoQ [3] also attempted to utilize self- and cross-attention mechanisms to aggregate useful information, yet it still introduced an extra aggregator that includes encoder blocks and cross-attention layers, as well as a

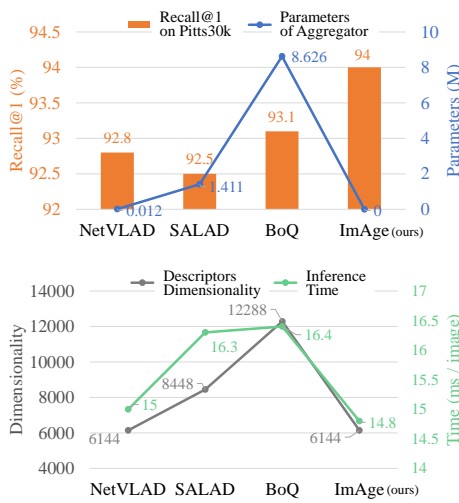

Figure 1: Comparison of three explicit aggregation methods and our ImAge. All methods use DINOv2-base-register as the backbone and are trained on the GSV-Cities dataset. ImAge achieves the best Recall@1 with the smallest descriptor dimension and the lowest inference time. Meanwhile, there is no extra explicit aggregator in our ImAge model.

large number of learnable queries. Another study [18] indicated that simply adding some registers, similar to concatenating the class token with patch tokens, can buffer excess global information (so-called "undesirable artifacts") into these registers. Unfortunately, it discarded these registers finally and lacked deeper research on the use of global information on registers.

In view of the issues of the previous explicit aggregation paradigm and the potential implicit aggregation ability of the transformer backbone itself, in this paper, we systematically explore the **Im**plicit **Agg**regation (abbreviated as ImAge) method, *i.e.*, unify feature extraction and aggregation solely via the backbone for VPR. Specifically, we introduce some learnable aggregation tokens, which are prepended to the patch tokens before a particular transformer encoder block. All these tokens will be jointly processed by the subsequent blocks and interact via the intrinsic self-attention mechanism, thus transmitting useful information within the patch tokens to our aggregation tokens. Finally, we only take aggregation tokens from the output of the last block and concatenate them to serve as the global descriptor, thereby achieving implicit aggregation. The proposed VPR paradigm provides a novel perspective different from the previous paradigm, unifying feature extraction and aggregation into a more cohesive framework. This further enables progressive aggregation in cascaded transformer blocks (rather than one-shot aggregation by a separate aggregator), thus achieving the correction and refinement of global image representations (*i.e.*, our aggregation tokens). Moreover, where and how to add aggregation tokens, as well as the initialization of these tokens, significantly impact performance. To this end, we propose an optimal token insertion strategy and token initialization method to effectively and efficiently yield more robust image representations and thus achieve excellent VPR performance. Our ImAge brings the following **contributions**:

1) We propose an implicit aggregation method to produce robust VPR image representations, which neither modifies the backbone nor needs an extra aggregator. It only adds some aggregation tokens before a specific block of the transformer backbone, leveraging the inherent self-attention mechanism to implicitly aggregate patch features. Our method provides a novel perspective different from the previous paradigm, effectively and efficiently achieving better performance in the transformer era.

2) To further improve the performance and efficiency of our ImAge, we propose: a) an aggregation token insertion strategy that deliberately delays token insertion until a specific transformer block, where patch tokens have acquired sufficient representation capability; b) a token initialization method that uses the L2-normalized cluster centers yielded by the $k$-means method to initialize added tokens.

3) Extensive experiments show that our ImAge significantly outperforms the latest explicit aggregation methods (*e.g.*, SALAD and BoQ) with the same setup (see Fig. 1). Besides, our method also achieves state-of-the-art (SOTA) results (*e.g.*, ranks 1st on MSLS challenge leaderboard) with high efficiency.

## 2   Related Work

**Visual Place Recognition:** Early research on VPR primarily focused on aggregating the hand-crafted descriptors [7] to global descriptors using some classical aggregation algorithms, such as Bags of Words [60] and VLAD [35, 45, 64, 5, 41]. In light of the remarkable achievements of deep learning across numerous computer vision tasks, contemporary VPR approaches [63, 4, 38, 17, 54, 26, 27, 69, 70, 28, 11, 44, 22] have increasingly utilized diverse deep features for better performance. Besides, traditional aggregation algorithms are gradually replaced by trainable aggregation layers, *e.g.*, NetVLAD [4] and GeM pooling [56]. Although some methods [30, 9, 59, 47, 50] employ local feature matching for re-ranking after initial global feature retrieval to boost performance, the backbone-plus-aggregator paradigm has been the de-facto standard [10] in VPR over the past decade. Some recent research [8, 1, 2, 34, 3, 49] has proposed several alternative approaches following this paradigm. For instance, CricaVPR [49] leveraged a cross-image encoder to produce cross-image correlation-aware global representations. SALAD [34] redefined the soft assignment in NetVLAD as an optimal transport problem and used the Sinkhorn algorithm to solve it. BoQ [3] employed distinct learnable queries to probe the input features through cross-attention, facilitating better information aggregation. These methods achieved excellent results using the ViT-based foundation model DINOv2 as the backbone. Unlike these methods that meticulously design auxiliary aggregators to yield global features, our ImAge method presents a novel paradigm that only introduces some additional tokens to the transformer backbone to conduct implicit aggregation via the inherent self-attention mechanism in transformers, thus achieving a simpler architecture and more powerful performance.

**Additional Tokens in Transformers:** Popularized by BERT [20], integrating special tokens into the token sequence in transformers has been a promising design choice for various purposes. We group such extra tokens into 3 categories based on their functional roles. **1)** *Output-oriented tokens* are learnable anchors that collect information from patch tokens, whose output values are then transmitted as task-specific outputs, *e.g.*, the class tokens used in BERT [20] and ViT [23] for classification, as well as detection tokens in YOLOS [25] for object detection. **2)** *Prompt tokens* act as trainable continuous vectors that replace traditional discrete text prompts, efficiently guiding pretrained transformer models to adapt to specific tasks by adjusting the model input, without modifying the parameters of models [43, 42, 36], which has become an essential branch of parameter-efficient fine-tuning methods [32]. **3)** *Memory tokens* act as registers that hold intermediate states during sequential processing steps, tracing their roots to neural memory architectures [14, 13]. This approach gains critical support from the DINOv2-register work [18], which observed that vision transformers improperly re-purpose background patch tokens as implicit memories when the standard class token lacks the capacity to accommodate global semantics. To address this, they prepend multiple memory tokens called registers to input tokens, which provide extra storage for buffering of global context, thus eliminating artifacts. Inspired by this work, we introduce the concept of **aggregation tokens** to effectively absorb global context from patch tokens. However, register tokens are discarded from the final output after temporary use, contrasting with our aggregation tokens that directly form the output descriptor for VPR (*i.e.*, our method falls into the "output-oriented tokens" category). Among VPR methods, BoQ [3] also advocated the introduction of a bag of output-oriented tokens named queries for aggregation (but in the aggregator rather than backbone). While effective, BoQ uses extra encoder blocks and cross-attention layers as the aggregator. In contrast, our method directly employs the inherent self-attention mechanism of the backbone, offering unique advantages.

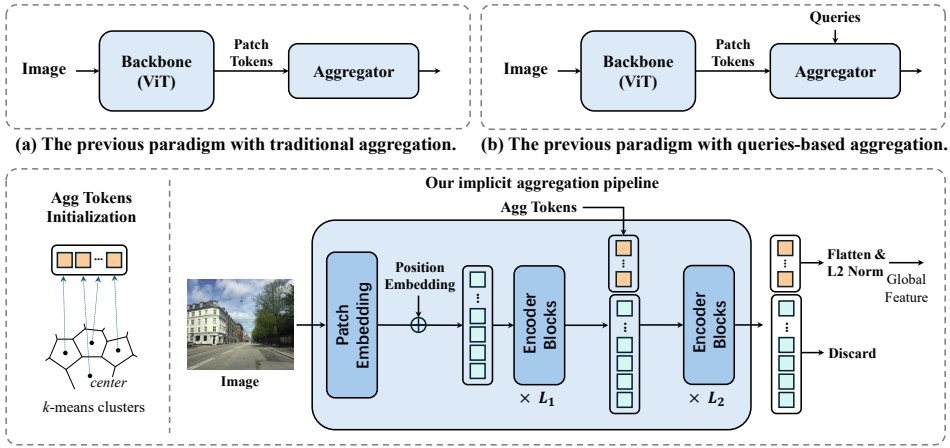

**(c) Our implicit aggregation paradigm (without explicit aggregator).**

Figure 2: Illustration of the previous paradigm and our ImAge paradigm. (a) The backbone-plus-aggregator paradigm with the traditional aggregator. (b) The backbone-plus-aggregator paradigm with a queries-based aggregator that introduces some queries to learn global information from the patch tokens. (c) Our ImAge only prepends a set of aggregation tokens to the patch tokens before a specific block in transformer backbone, making them interact globally via self-attention to achieve implicit aggregation. Notably, these aggregation tokens are simply initialized by the $k$-means algorithm.

## 3 Methodology

This section begins with a review of the ViT [23] and the self-attention mechanism in it, which serves as the foundation for our ImAge method. Following that, we first present the pipeline of our method. Then, we introduce the insertion strategy of our aggregation tokens and their initialization method.

### 3.1 Preliminary

ViT [23] and its variants have rapidly emerged as the preferred backbones for a variety of computer vision tasks [71, 74, 73, 72, 50], owing to their exceptional capacity for modeling global relationships [58]. Given an input image of size $H \times W$, ViT partitions it into $N = HW/P^2$ non-overlapping patches. Each patch is then flattened and linearly projected to create a $D$-dimensional token $x_p^i$. A learnable class token $x_{\mathsf{CLS}} \in \mathbb{R}^D$ is prepended to this sequence, and positional embeddings are added to encode spatial information, forming the initial input token sequence $z_0 = [x_{\mathsf{CLS}}, x_p^1, \ldots, x_p^N] \in \mathbb{R}^{(N+1) \times D}$. This sequence is iteratively processed through $L$ transformer encoder blocks. Each block comprises three core components: layer normalization (LN), multi-head self-attention (MHSA), and multi-layer perceptron (MLP). The $l$-th block updates the input $z_{l-1}$ to $z_l$ via

$$
\begin{aligned}
z_l' &= \mathrm{MHSA}\big(\mathrm{LN}(z_{l-1})\big) + z_{l-1}, \\
z_l &= \mathrm{MLP}\big(\mathrm{LN}(z_l')\big) + z_l'.
\end{aligned}
\tag{1}
$$

Within the MHSA module, the input sequence undergoes parallel linear transformations to generate $h$ independent sets of queries $Q$, keys $K$, and values $V$, each parameterized by learnable projection matrices. For each attention head, the scaled dot-product attention

$$
\mathrm{Attn}(Q, K, V) = \mathrm{Softmax}\big(QK^\top / \sqrt{d}\big) V, \quad d = D/h,
\tag{2}
$$

computes context-aware similarity scores and dynamically aggregates information across all tokens. This mechanism facilitates rich cross-token interactions, where each token selectively assimilates features from others based on pairwise affinities. The outputs of all heads are concatenated to integrate multi-subspace representations and then linearly projected again, synthesizing position-wise updated embeddings $z_l'$ that encode global contextual relationships. These properties of ViT indicate its potential to aggregate patch tokens by introducing additional tokens, which we will introduce below.

## 3.2 Implicit Aggregation via the Transformer Backbone

After extracting the patch features/tokens via the backbone, there are primarily two ways in previous works to obtain robust global descriptors. One is to directly aggregate these patch tokens with a common aggregator (*e.g.*, NetVLAD [4] and SALAD [34]), as in Fig. 2 (a). The other uses the queries-based aggregator to learn global information from the patch tokens (*e.g.*, BoQ [3] and EDTformer [37]), as in Fig. 2 (b). However, our ImAge will essentially eliminate the use of aggregators.

An overview of our ImAge is presented in Fig. 2 (c). Unlike existing VPR methods, ImAge removes the explicit aggregator and uses only the backbone network to achieve implicit feature aggregation. In this work, we utilize the vision transformer as the backbone, making the first $L_1$ encoder blocks process the patch tokens as usual. After these encoder blocks, a set of $M$ learnable aggregation (agg) tokens, formulated as $a \in \mathbb{R}^{M \times D}$, is introduced and prepended to the other tokens $z$, getting a new sequence $[a, z]$. Then, these combined tokens will be uniformly processed by the subsequent $L_2$ encoder blocks and perform global interactions via the internal self-attention mechanism. Specifically, $[a, z]$ is first linearly transformed to produce the query $Q = [Q_a, Q_z]$, key $K = [K_a, K_z]$, and value $V = [V_a, V_z]$. Next, the interactions are computed according to Eq. 2 as follows:

$$Attn(Q, K, V) = [Q_a, Q_z][K_a, K_z]^\top[V_a, V_z] = [\underbrace{Q_a K_a^\top V_a}_{\text{Agg-Agg}} + \underbrace{Q_a K_z^\top V_z}_{\text{Agg-Patch}}, Q_z K_a^\top V_a + Q_z K_z^\top V_z],$$

(3)

where we omit the Softmax and scaling operations for simplicity. Based on Eq. 3, it is evident that the self-attention layers within the backbone enable us to achieve two key objectives: 1) Agg tokens can focus on their own features by agg-agg attention, thereby enhancing their intrinsic representation capabilities; 2) More importantly, agg tokens can fully learn and capture the global contextual information within the patch tokens by agg-patch attention, thus achieving robust implicit aggregation. Finally, we take the agg tokens from the output of the last encoder block, which are flattened into a vector and L2-normalized to form the final global image representation. It is worth noting that in the previous backbone-plus-aggregator paradigm, the global image representation is formed after one-shot aggregation of patch features implemented by the aggregator and is immediately output (without opportunity for refinement). Our method, however, adds agg tokens before a specific block of the transformer backbone. These agg tokens serve as global representations, and they are subsequently corrected and refined in subsequent blocks (synchronously with the refinement of patch tokens), rather than being aggregated/yielded all at once. This is an advantage over the previous paradigm.

Obviously, our ImAge fundamentally diverges from the practices of prompt tuning (aim to fine-tune models) [42] and register tokens (aim to remove artifacts) [18], which discard the newly added tokens finally. Besides, our method also differs from the class token. Our agg tokens have better scalability, along with different insertion strategies and initialization methods, which will be described below.

## 3.3 The Insertion Strategy of Aggregation Tokens

Our implicit aggregation method provides a robust image representation for VPR in an extremely simple manner. It requires neither explicit aggregators nor any modifications to the original backbone. However, where and how to add our agg tokens remains an open issue worthy of further exploration. For instance, previous works such as prompt tuning and DINOv2-register prepend additional tokens to the patch tokens (and class token) before the first transformer block, as

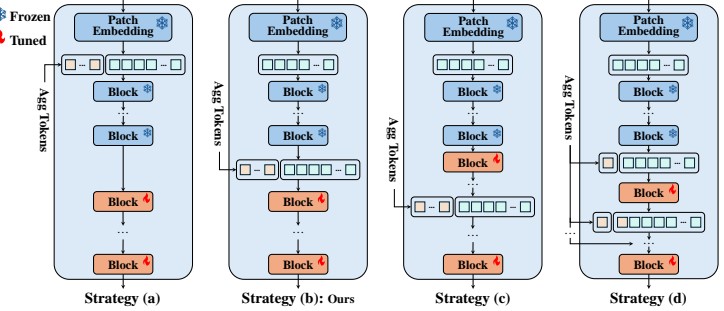

Figure 3: Illustration of 4 insertion strategies for agg tokens. (a) Agg tokens are added before all transformer blocks. (b) Agg tokens are added at the junction between frozen and trainable blocks (our strategy). (c) Agg tokens are added at a deeper tunable block. (d) Agg tokens are added incrementally across multiple blocks rather than all at once.

shown in Fig. 3 (a). Our objective differs from these works, and we no longer follow this way regarding the specific placement of agg tokens. More specifically, there are two reasons for this: 1) Our goal is to aggregate patch tokens with meaningful representations. Since early transformer blocks produce relatively weak features, adding agg tokens at the beginning is unnecessary and potentially detrimental to model performance. 2) In the field of VPR, the common practice for model training is to fine-tune only the last few blocks (layers) of the pre-trained model on the VPR dataset, while freezing the previous blocks. If agg tokens are added at the beginning, although most of the shallow and middle blocks are frozen, the added agg tokens need to be tuned. According to the chain rule of back-propagation [52], the gradients of the parameters in these frozen blocks still need to be calculated, leading to significant GPU memory and computational overhead.

In light of the above considerations, our strategy is to prepend agg tokens only when the patch tokens have acquired sufficient representational capability. A more specific criterion is to add the agg tokens at the junction between frozen and trainable transformer blocks, as illustrated in Fig. 3 (b). For example, in the case of the DINOv2 backbone, most previous VPR methods [34, 51] only fine-tune the last four blocks. Accordingly, we prepend the agg tokens to the patch tokens before the fourth-to-last block. Since the preceding blocks are frozen, it indicates that the features output here are general enough. The subsequent blocks are then trained on the VPR dataset to produce features more suitable for the VPR task, so our agg tokens can also learn better task-specific global representations. Additionally, we consider two alternative strategies. One is to add agg tokens before a deeper trainable block, as shown in Fig. 3 (c). The other is to add agg tokens progressively instead of all at once, as shown in Fig. 3 (d). However, both ways reduce the opportunities for the refinement and correction of image representations, which leads to suboptimal performance. Based on these objective factors, we finally propose the aforementioned strategy (b) for the insertion of agg tokens.

### 3.4   The Initialization of Aggregation Tokens

The agg tokens are learnable parameters, and their initialization can significantly impact the model performance. Prior to training on VPR datasets, the model is typically pre-trained on large-scale datasets. As a result, the patch tokens output by the specific block of such a model already have good representational capabilities. If our agg token is inappropriately initialized and prepended to patch tokens, it will instead cause damage to the representation of patch tokens in the subsequent processing of the MHSA layer. So, proper initialization of the agg token is essential.

Fortunately, a similar issue has been discussed in NetVLAD [6]. This method determines $k$ cluster centers (and the parameters of the assignment layer) through training. The residual statistics from patch features to cluster centers are used as the global representation. At the beginning, it also requires initializing $k$ cluster centers and the soft-assignment layer through the unsupervised $k$-means algorithm to achieve good performance. Although our ImAge method uses the self-attention mechanism to perform implicit aggregation, its essence can be regarded as each added agg token representing a unique category (but not necessarily corresponding to an object category in human semantics, such as building or vegetation) that is helpful to VPR, similar to each cluster in NetVLAD. Therefore, we can learn from NetVLAD, using the $k$-means algorithm to perform unsupervised clustering for the initialization of agg tokens. Besides, NetVLAD uses L2-normalized cluster centers to initialize parameters (weight $\mathbf{w}$) in the assignment layer. Through our empirical research, the L2-normalized centers can reduce the impact of extreme cases and are more suitable for initializing agg tokens than the original centers, *i.e.*, it is our final method.

## 4   Experiments

### 4.1   Datasets and Performance Evaluation

**Datasets.** We conduct the experiments on several VPR benchmark datasets. These datasets exhibit various challenges, including viewpoint changes, condition variations, and the perceptual aliasing issue. Table 1 provides a summary of the main evaluation datasets. **MSLS** [68] is a particularly challenging dataset, in which images are taken from urban, suburban, and natural scenes, covering diverse visual changes. **Pitts30k** [65] extracted from Google Street View, mainly presents severe variations in viewpoint. **Tokyo24/7** [64] shows dramatic condition (light) changes. **Nordland** [62] is gathered across four seasons with a fixed perspective from the front of a train. Moreover, we also use

the Baidu Mall [61], SPED [16], Pitts250k [65], St. Lucia [29], and SVOX [12] datasets for a few supplementary experiments.

**Performance Evaluation.** We follow the previous work [8, 10] using the Recall@N (R@N) as the evaluation metric for recognition performance. R@N is the proportion of queries for which at least one of the top-N predicted images is within a threshold of ground truth. We set the threshold to 10 frames for Nordland and 25 meters for others, as in this benchmark [10].

Table 1: Summary of the main evaluation datasets.

| Dataset | Description | Number | |
|---|---|---|---|
| | | Database | Queries |
| Pitts30k | urban, panorama | 10,000 | 6,816 |
| MSLS-val | urban, suburban | 18,871 | 740 |
| MSLS-challenge | long-term | 38,770 | 27,092 |
| Tokyo24/7 | urban, day/night | 75,984 | 315 |
| Nordland | natural, seasonal | 27,592 | 27,592 |

### 4.2 Implementation Details

The experiments are conducted on the NVIDIA RTX A6000 GPU using PyTorch. We use DINOv2-base-register as the backbone and only fine-tune the last four transformer blocks with the previous layers frozen. The token dimension in backbone is 768, and the number of our aggregation tokens is 8, thus outputting 6144-dim global descriptors. The image resolution is 224×224 for training and 322×322 for inference, as in SALAD [34]. We employ the multi-similarity loss [67] for training, with hyperparameters set following the GSV-Cities work [1]. The model is trained using the Adam optimizer with an initial learning rate of 0.00005, halved every 3 epochs. Each training batch contains 120 places, with 4 images per place (*i.e.*, 480 images). Besides, we set the maximum epochs to 20.

### 4.3 Comparisons with State-of-the-Art Methods

This section shows the experimental comparison of our ImAge with SOTA methods, including 11 single-stage VPR methods: NetVLAD [4], SFRS [28], CosPlace [8], MixVPR [2], EigenPlaces [11], CricaVPR [49], SALAD [34], SALAD-CM [33], BoQ [3], SuperVLAD [51] and EDTformer [37], as well as 2 two-stage VPR methods (TransVPR [66] and SelaVPR [50]) that leverage local features for re-ranking. The latest studies, CricaVPR, SALAD, SALAD-CM, BoQ, SelaVPR, SuperVLAD, and EDTformer, all use the foundation model DINOv2 as the backbone to extract deep features and achieve SOTA results. Our method mainly adopts DINOv2-base-register in experiments. Additionally, Cosplace and EigenPlaces construct an extra large-scale dataset (SF-XL) for training. CircaVPR, SALAD, BoQ, and EDTformer are trained on the GSV-Cities dataset, while SALAD-CM combines GSV-Cities and MSLS-train for training. Our work further merges Pitts30k-train, MSLS-train, SF-XL, and GSV-Cities for training, following the process in SelaVPR++ [48]. Table 2 presents the comprehensive quantitative results. Moreover, to enable a fairer comparison among three leading aggregation methods (NetVLAD, SALAD, and BoQ) and our ImAge, we conduct a consistent comparison using the same setup (backbone, training data, image resolution), as shown in Table 3. The experiments using other transformer backbones (ViT and CLIP) are shown in Appendix D.

**For the comprehensive comparison in Table 2**: Compared to existing SOTA methods (*e.g.*, SALAD-CM, BoQ, and EDTformer), our ImAge removes the explicit aggregator and only uses the backbone to obtain robust global descriptors, thus achieving a promising performance. On Pitts30k, a benchmark known for its extreme viewpoint variations, EDTformer and BoQ achieve 93.4% and 93.7% R@1, respectively. In comparison, our ImAge achieves a notable 94.1% R@1, attaining a new level. This indicates that global descriptors produced by our ImAge are highly robust to viewpoint changes. SALAD-CM significantly outperforms other methods on the MSLS dataset, which presents greater challenges due to diverse visual changes and perceptual aliasing. Nevertheless, our ImAge method further advances recognition performance, achieving 94.5% R@1 on MSLS-val and 93.8% R@5 on MSLS-challenge (ranks 1st on the official leaderboard). On Tokyo24/7, which is characterized by severe illumination changes, our ImAge also achieves the best performance with 97.1% R@1. In addition to its competitive performance on urban and suburban datasets, our ImAge still performs well on natural image datasets suffering from seasonal variations. Specifically, ImAge achieves an almost perfect R@5 (*i.e.*, > 99.0%) on Nordland. Overall, compared with other SOTA methods, our ImAge delivers substantial performance improvements across diverse scenarios. More importantly, our method no longer relies on a dedicated aggregator to obtain such robust global features.

**For the fairer comparison in Table 3:** In this comparison, we use the same training dataset (GSV-Cities), backbone (DINOv2-base-register), and input image resolution (224×224 in training and

Table 2: Comprehensive comparison to existing SOTA VPR methods on multiple benchmark datasets. All methods follow the settings of their respective original works, so there are differences in the backbone, training set, image resolution, etc. The best results are highlighted in **bold** and the second are underlined. † CricaVPR and SuperVLAD use a cross-image encoder to correlate multiple images from the same place to achieve better performance on Pitts30k. They are not included in the comparison with others (on all datasets).

| Method | Dim | Pitts30k | | | MSLS-val | | | MSLS-challenge | | | Tokyo24/7 | | | Nordland | | |
|---|---|---|---|---|---|---|---|---|---|---|---|---|---|---|---|---|
| | | R@1 | R@5 | R@10 | R@1 | R@5 | R@10 | R@1 | R@5 | R@10 | R@1 | R@5 | R@10 | R@1 | R@5 | R@10 |
| NetVLAD [4] | 32768 | 81.9 | 91.2 | 93.7 | 53.1 | 66.5 | 71.1 | 35.1 | 47.4 | 51.7 | 60.6 | 68.9 | 74.6 | 6.4 | 10.1 | 12.5 |
| SFRS [28] | 4096 | 89.4 | 94.7 | 95.9 | 69.2 | 80.3 | 83.1 | 41.6 | 52.0 | 56.3 | 81.0 | 88.3 | 92.4 | 16.1 | 23.9 | 28.4 |
| TransVPR [66] | / | 89.0 | 94.9 | 96.2 | 86.8 | 91.2 | 92.4 | 63.9 | 74.0 | 77.5 | 79.0 | 82.2 | 85.1 | 63.5 | 68.5 | 70.2 |
| CosPlace [8] | 512 | 88.4 | 94.5 | 95.7 | 82.8 | 89.7 | 92.0 | 61.4 | 72.0 | 76.6 | 81.9 | 90.2 | 92.7 | 58.5 | 73.7 | 79.4 |
| MixVPR [2] | 4096 | 91.5 | 95.5 | 96.3 | 88.0 | 92.7 | 94.6 | 64.0 | 75.9 | 80.6 | 85.1 | 91.7 | 94.3 | 76.2 | 86.9 | 90.3 |
| EigenPlaces [11] | 2048 | 92.5 | 96.8 | 97.6 | 89.1 | 93.8 | 95.0 | 67.4 | 77.1 | 81.7 | 93.0 | 96.2 | 97.5 | 71.2 | 83.8 | 88.1 |
| SelaVPR [50] | / | 92.8 | 96.8 | 97.7 | 90.8 | 96.4 | 97.2 | 73.5 | 87.5 | 90.6 | 94.0 | 96.8 | 97.5 | 87.3 | 93.8 | 95.6 |
| CricaVPR† [49] | 4096 | 94.9† | 97.3† | 98.2† | 90.0 | 95.4 | 96.4 | 69.0 | 82.1 | 85.7 | 93.0 | 97.5 | 98.1 | 90.7 | 96.3 | 97.6 |
| SuperVLAD† [51] | 3072 | 95.0† | 97.4† | 98.2† | 92.2 | 96.6 | 97.4 | 75.3 | 86.8 | 89.9 | 95.2 | 97.8 | 98.1 | 91.0 | 96.4 | 97.7 |
| SALAD [34] | 8448 | 92.5 | 96.4 | 97.5 | 92.2 | 96.4 | 97.0 | 75.0 | 88.8 | 91.3 | 94.6 | 97.5 | 97.8 | 89.7 | 95.5 | 97.0 |
| SALAD-CM [33] | 8448 | 92.7 | 96.8 | 97.9 | 94.2 | 97.2 | 97.4 | 82.7 | 91.2 | 92.7 | 96.8 | 97.5 | 97.8 | 96.0 | 98.5 | 99.2 |
| BoQ [3] | 12288 | 93.7 | 97.1 | 97.9 | 93.8 | 96.8 | 97.0 | 79.0 | 90.3 | 92.0 | 96.5 | 97.8 | **98.4** | 90.6 | 96.0 | 97.5 |
| EDTformer [37] | 4096 | 93.4 | 97.0 | 97.9 | 92.0 | 96.6 | 97.2 | 78.4 | 89.8 | 91.9 | **97.1** | **98.1** | **98.4** | 88.3 | 95.3 | 97.0 |
| ImAge (Ours) | 6144 | **94.1** | **97.3** | **98.1** | **94.5** | **97.3** | **98.0** | **84.5** | **93.8** | **95.4** | **97.1** | **98.1** | **98.4** | **97.7** | **99.3** | **99.6** |

Table 3: Consistent comparison to SOTA VPR aggregation algorithms. *All methods consistently use the same backbone (DINOv2-base-register), training dataset (GSV-Cities), and image resolution.

| Method | Dim | Param. in Aggre. | Inference Time (ms) | Pitts30k | | | MSLS-val | | | Tokyo24/7 | | | Nordland | | |
|---|---|---|---|---|---|---|---|---|---|---|---|---|---|---|---|
| | | | | R@1 | R@5 | R@10 | R@1 | R@5 | R@10 | R@1 | R@5 | R@10 | R@1 | R@5 | R@10 |
| NetVLAD* | 6144 | 0.012 M | 15.0 | 92.8 | 96.6 | 97.8 | 91.8 | 96.5 | 96.6 | 95.6 | **98.1** | 98.7 | 90.5 | 96.5 | 97.8 |
| SALAD* | 8448 | 1.411 M | 16.3 | 92.5 | 96.6 | 97.5 | 92.6 | 96.6 | 97.0 | 95.6 | 97.5 | **99.0** | 86.5 | 93.6 | 95.7 |
| BoQ* | 12288 | 8.626 M | 16.4 | 93.1 | **97.2** | **98.0** | 92.8 | 96.5 | 97.0 | 95.2 | 97.7 | 98.2 | 87.0 | 94.0 | 95.9 |
| ImAge* | 6144 | 0 M | 14.8 | **94.0** | **97.2** | **98.0** | **93.0** | **97.0** | **97.2** | **96.2** | **98.1** | 98.4 | **93.2** | **97.6** | **98.6** |

Table 4: Consistent comparison to SOTA VPR aggregation algorithms on supplementary datasets. *All methods consistently use the same backbone (DINOv2-base-register), training dataset (GSV-Cities), and image resolution.

| Method | Dim | Baidu Mall | | SPED | | Pitts250k | | St. Lucia | | SVOX-Night | | SVOX-Sun | |
|---|---|---|---|---|---|---|---|---|---|---|---|---|---|
| | | R@1 | R@5 | R@1 | R@5 | R@1 | R@5 | R@1 | R@5 | R@1 | R@5 | R@1 | R@5 |
| NetVLAD* | 6144 | 69.8 | 82.5 | 91.1 | 94.9 | 95.6 | 98.5 | 99.9 | 99.9 | 97.0 | 98.9 | 97.7 | 99.2 |
| SALAD* | 8448 | 67.3 | 81.2 | 90.3 | 94.6 | 95.4 | 98.8 | 99.9 | 100 | 96.1 | 99.0 | 97.2 | 99.4 |
| BoQ* | 12288 | 65.6 | 79.2 | 90.3 | **96.0** | 95.6 | 98.9 | 99.9 | 100 | 97.4 | **99.5** | 97.4 | 99.3 |
| ImAge* | 6144 | **70.6** | **83.8** | **91.6** | 95.6 | **96.5** | **99.1** | 99.9 | 100 | **97.6** | 99.4 | **98.0** | **99.5** |

322×322 in inference) for all methods. It is worth mentioning that Fig. 1 has shown some of the results of Table 3. In summary, our ImAge achieves the best overall performance on all datasets with the smallest descriptor dimension, the fastest inference speed, and the fewest model parameters. Note that even considering the additional parameters brought by our agg tokens, it is only 0.006M, *i.e.*, half of NetVLAD (0.07% of BoQ). This further supports our statement that an elaborately designed aggregator is not indispensable in the transformer era for robust global descriptors.

Besides, we also conduct the consistent comparison experiments on some supplementary datasets, including Baidu Mall [61], SPED [16], Pitts250k [65], St. Lucia [29], and SVOX [12], and the results are shown in Table 4. Compared to the three SOTA explicit aggregation methods, our ImAge achieves the best R@1 performance on all supplementary datasets. In particular, on Baidu Mall, which is the only indoor dataset and exhibits a distinct visual distribution from the other outdoor datasets, our method achieves the best performance, outperforming NetVLAD, SALAD, and BoQ with 0.8%, 3.3%, and 5.0% absolute R@1 improvements, respectively. This demonstrates that the global descriptors produced by our ImAge method through implicit aggregation are not only highly robust against common visual changes but also exhibit superior generalization ability.

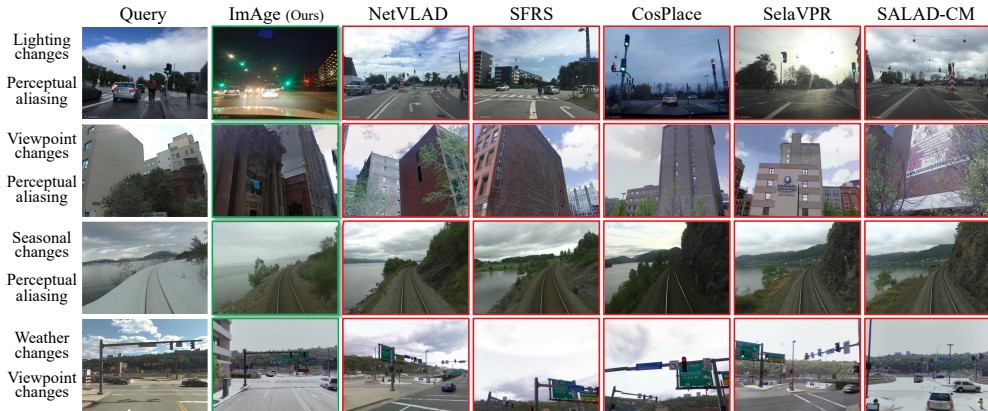

Figure 4: Qualitative results. In these four challenging scenarios (involving dynamic objects, severe viewpoint variations, condition changes, etc.), our proposed ImAge method consistently retrieves the correct results from the database, while other methods all return the wrong images.

Fig. 4 presents qualitative retrieval results, where the proposed ImAge consistently demonstrates high robustness in various extreme scenes. For example, the first three cases exhibit severe lighting changes, viewpoint variants, and seasonal transitions, respectively. Other methods often retrieve visually similar but actually incorrect results due to perceptual aliasing. However, our ImAge effectively addresses these challenges in VPR and successfully returns the right results.

## 4.4 Ablation Studies

In this section, we conduct a series of ablation studies on our ImAge. We uniformly use the DINOv2-base-register backbone and train models on GSV-Cities with the batch size set to 120 (as the experiment in Table 3). Unless stated otherwise, we only fine-tune the last four transformer blocks.

**Effect of tokens insertion strategy.** In Section 3.3, we discussed several strategies for adding agg tokens and proposed the optimal strategy. To validate its effectiveness, we conduct an ablation to compare different strategies. To be fair, we consistently add 8 agg tokens. Strategy (a) and (â) both add agg tokens before the first transformer block. The only difference is that all transformer blocks in (â) are trainable. Strategy (b) is our optimal strategy. Strategy (c) introduces agg tokens before the penultimate block. Strategy (d) progressively adds 2 agg tokens before each of the last four blocks. Results are presented in Table 5. Among them, (a) performs the worst, because the early frozen transformer blocks produce weak and less informative features for VPR, harming the agg tokens to effectively capture meaningful global information. The issue is mitigated in (â), which further confirms our hypothesis (*i.e.*, adding agg tokens before the first trainable blocks). However, (â) trains all blocks, which incurs a lot of computational overhead and damages the excellent transferability of foundation models, thus failing to get optimal results. When fine-tuning only the last four transformer blocks, our proposed strategy (b) consistently outperforms all alternatives on all datasets by a large margin. This is because the last four tunable blocks can produce more suitable features for the VPR task, so our agg tokens can fully learn task-specific global representations. Although (c) and (d) also show relatively competitive performance, the late or gradual addition of agg tokens provides fewer opportunities to interact with patch features, thus limiting their ability to learn better representations.

**Effect of aggregation tokens initialization.** To validate the effectiveness of our proposed initialization methods for agg tokens, we conduct an ablation study using four initialization strategies: zero initialization (*i.e.*, no initialization), normal distribution initialization (commonly used for the class token or register tokens initialization [18]), vanilla cluster centers (yielded by $k$-means) initialization, and L2-normalized cluster centers initialization (*i.e.*, ours). We consistently use 8 agg tokens and prepend them to the patch tokens before the fourth-to-last transformer block. The experimental results are presented in Table 6. Zero initialization produces uniform representations at the beginning, limiting (even harming) interaction between agg tokens and patch features, and hindering global context modeling. In contrast, normal initialization provides a better inductive bias during early training and introduces slight randomness into the agg tokens, which helps break symmetry to get better performance. However, both initialization methods lack any visual prior, forcing the agg tokens

Table 5: Comparison of different insertion strategies for agg tokens. The **strategy (b)** is ours.

| Method | Pitts30k | | MSLS-val | | Nordland | |
|---|---|---|---|---|---|---|
| | R@1 | R@5 | R@1 | R@5 | R@1 | R@5 |
| Strategy (a) | 88.5 | 94.1 | 83.6 | 90.9 | 40.4 | 56.2 |
| Strategy (â) | 92.6 | 96.9 | 92.0 | 96.6 | 89.0 | 95.6 |
| **Strategy (b)** | **94.0** | **97.2** | **93.0** | **97.0** | **93.2** | **97.6** |
| Strategy (c) | 93.2 | 97.1 | 92.2 | 96.5 | 88.1 | 95.0 |
| Strategy (d) | 93.3 | 97.1 | 92.4 | 96.6 | 90.3 | 96.4 |

Table 6: Comparison of different initializations for agg tokens. The **centers-L2N** is ours.

| Method | Pitts30k | | MSLS-val | | Nordland | |
|---|---|---|---|---|---|---|
| | R@1 | R@5 | R@1 | R@5 | R@1 | R@5 |
| zero | 92.1 | 96.6 | 89.6 | 95.1 | 68.9 | 82.7 |
| normal_distrib | 92.9 | 96.9 | 92.0 | 96.8 | 88.6 | 95.3 |
| centers | 93.5 | 96.9 | 92.6 | 96.9 | 91.7 | 97.0 |
| **centers-L2N** | **94.0** | **97.2** | **93.0** | **97.0** | **93.2** | **97.6** |

to learn the patterns relevant to VPR from scratch, which constrains their final performance. Initializing agg tokens with cluster centers can be viewed as injecting a data-driven prior. These centers, obtained via unsupervised clustering of descriptors from randomly sampled training images, tend to capture common visual patterns. Such initialization can facilitate agg tokens to learn meaningful global information and diminish useless elements. Moreover, L2-normalized cluster centers offer more robust initializations for agg tokens by mitigating the influence of outliers, thereby achieving the optimal performance on all datasets.

**Effect of the number of aggregation tokens.**
In this subsection, we investigate the impact of the number of added agg tokens (and use the class token, *i.e.*, cls, as baseline). The agg tokens are all added before the fourth-to-last block, and the results are in Table 7. Even with a single agg token, ImAge demonstrates a clear advantage over the class token with the same dimensionality, notably achieving an 11.3% absolute R@1 improvement on Nordland. This proves the differences and advantages of our method compared with directly using the class token, as

Table 7: Comparison with the ImAge ablated versions with different numbers of aggregation tokens.

| Number | Pitts30k | | MSLS-val | | Nordland | |
|---|---|---|---|---|---|---|
| | R@1 | R@5 | R@1 | R@5 | R@1 | R@5 |
| cls | 91.8 | 96.5 | 89.1 | 95.3 | 63.5 | 79.0 |
| 1 | 92.2 | 96.6 | 90.7 | 95.4 | 74.8 | 87.0 |
| 4 | 93.4 | 97.0 | 92.3 | 96.6 | 89.6 | 96.1 |
| 8 | **94.0** | **97.2** | **93.0** | **97.0** | **93.2** | **97.6** |
| 16 | 93.7 | **97.2** | 92.8 | 96.9 | 92.2 | 97.2 |
| 32 | 93.1 | 96.9 | 92.6 | 96.8 | 90.3 | 96.2 |
| 64 | 92.8 | 96.8 | 92.2 | 96.5 | 85.4 | 93.0 |

well as the excellent performance of our method with low-dimensional descriptors. Furthermore, performance consistently improves as the number of agg tokens increases, with the best results obtained using 8 agg tokens. This is because a moderate increase of agg tokens enables more sufficient interaction and finer aggregation from the patch tokens. However, when the number of agg tokens becomes excessively large (*e.g.*, 64), a noticeable decline is observed. This may be attributed to the global nature of self-attention, where an excessive number of agg tokens can interfere with the contextual information of patch tokens, thereby indirectly degrading their own representational capability. Thus, adding 8 agg tokens is a promising choice overall.

## 5 Conclusions

In this paper, we presented ImAge, an innovative paradigm that explores implicit aggregation with a transformer to produce robust global image representation for VPR. Our method only adds some aggregation tokens and leverages the inherent self-attention of the transformer backbone to implicitly aggregate patch features. It overcomes the limitations of the previous backbone-plus-aggregator paradigm in an extremely simple manner, which neither modifies the original backbone nor requires an extra aggregator. Moreover, we propose an aggregation token insertion strategy and a token initialization method for our ImAge method to further improve the performance and efficiency. Experimental results show that ImAge obviously outperforms the latest explicit aggregation methods with higher efficiency under the same setup and achieves SOTA results on common VPR datasets.

## Acknowledgments and Disclosure of Funding

This work was supported by the National Key R&D Program of China (2022YFB4701400/4701402), SSTIC Grant (KJZD20230923115106012, KJZD20230923114916032, GJHZ20240218113604008), National Natural Science Foundation of China (62402252, 62536003), and Guangdong High-Level Talent Programme (2024TQ08X283).

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

## A    Broader Impacts

Visual place recognition plays an important role in several applications, including autonomous driving, augmented reality, and robot localization. Our work proposes an implicit aggregation method to produce robust image representation for VPR with the transformer-based backbone and shows SOTA performance. While our exploration on VPR remains fundamental and application-agnostic, the potential utilization of VPR technology for intrusive surveillance and social media monitoring raises some privacy issues. It is crucial to prevent the misuse of VPR research for detrimental purposes.

## B    Limitations and Future Work

While our study provides a novel insight (*i.e.*, implicit aggregation without an explicit aggregator) into achieving robust global image representation for VPR, we acknowledge three limitations of our work: **1)** Although our ImAge method demonstrates universality for the transformer-based models, compared to the foundation models pre-trained on the massive dataset (*e.g.*, DINOv2 pre-trained on the LVD-142M dataset [55]), the performance improvement is less pronounced when using backbones without pre-training on sufficiently massive data (*e.g.*, the ViT pre-trained only on ImageNet [19]). This will be shown in Appendix D. However, this also suggests that with the advancement of increasingly powerful foundation models, the superiority of our approach compared to existing VPR methods may become more prominent. **2)** The proposed method may not be a good choice when we want to keep the backbone frozen (like AnyLoc [39]) or when the backbone is extremely expensive to fine-tune. However, it is also worth noting that we only fine-tune the last few blocks of the transformer backbone in most cases, which is relatively cheap. **3)** Although this work focuses on the VPR task, we believe that the proposed ImAge is broadly applicable to a wide range of image (or other modalities) retrieval tasks. The potential of our ImAge method for more applications in the machine learning community needs to be further explored through more experiments in future work.

## C    More Details about the Relations & Differences to Other Methods

Our ImAge design draws inspiration from prior works, particularly DINOv2-register [18] and prompt tuning [42]. Both approaches introduce additional tokens to the transformer backbone before the first encoder block. However, their objectives and usage differ fundamentally from ours. Prompt Tuning aims to adapt a frozen model to downstream tasks by learning a set of prompt tokens in a parameter-efficient manner, while these tokens are typically excluded from the final representation. DINOv2-register introduces additional register tokens to mitigate artifacts in the feature maps. Although the study shows that these registers may capture certain global information, they are ultimately discarded, and only the patch and class tokens are used for downstream tasks. In contrast, our ImAge method introduces the aggregation (agg) tokens before a particular transformer block, and utilizes agg tokens from the output of the last block as the final global image representation, which provides a reverse perspective compared to these approaches. In addition, while the class token within the transformer backbone is sometimes used as a global representation, our ImAge differs in several key aspects and demonstrates superior scalability. First, the class token is typically introduced at the beginning of the transformer and starts to learn from shallow features, which may limit its flexibility and make it difficult to fully capture task-specific complex semantics. Second, the class token is a single fixed embedding, which inherently restricts its representational capacity. Although it may suffice for relatively simple classification tasks, it often proves inadequate for more complex scenarios requiring richer and more flexible representations. In contrast, our ImAge introduces agg tokens with customizable positions and quantities, allowing them to fully learn task-specific global features. Moreover, we also design a tokens initialization method based on the $k$-means algorithm, which is significantly different from other works. Additionally, among existing VPR methods, BoQ [3] also offers some valuable insights for our work. However, it elaborately designs an explicit aggregator consisting of additional encoder blocks and cross-attention layers, which aggregates global information from patch tokens into a set of extra learnable queries. In contrast, we focus on the transformer backbone itself and make use of the inherent self-attention mechanism. Our study reveals a new insight: the aggregation function, previously implemented through an exquisitely designed aggregator, already appears naturally in the transformer backbone. We demonstrate that, by adding just some additional tokens, we can fully develop this implicit and progressive aggregation behavior.

Table 8: Results of NetVLAD and our ImAge using CLIP (base version, only vision encoder) and ViT (base version) as backbone. All models are trained on the GSV-Cities dataset with the batch size equal to 120. The learning rate is 0.00006 for the CLIP-based model and 0.0003 for the ViT-based model. For ViT, the last two blocks are directly truncated and all other blocks are trainable, as in [10, 51]. For CLIP, we only train the last 6 blocks with the previous layers frozen. All methods produce 768*8-dimensional descriptors, *i.e.*, 8 clusters for NetVLAD and 8 aggregation tokens for ImAge, the same as in the main paper.

| Method | Pitts30k | | | MSLS-val | | | Nordland | | |
|---|---|---|---|---|---|---|---|---|---|
| | R@1 | R@5 | R@10 | R@1 | R@5 | R@10 | R@1 | R@5 | R@10 |
| CLIP-NetVLAD | 90.6 | **95.7** | **97.2** | 87.2 | **94.1** | 94.6 | 60.6 | **74.6** | **80.2** |
| CLIP-ImAge (Ours) | **91.2** | 95.6 | 96.9 | **88.2** | **94.1** | **95.5** | **61.0** | **74.6** | 80.1 |
| ViT-NetVLAD | 90.1 | 95.3 | 96.4 | 82.4 | 90.7 | 93.0 | 52.1 | 67.6 | 74.1 |
| ViT-ImAge (Ours) | **90.3** | **96.1** | **97.3** | **86.2** | **92.2** | **93.8** | **53.3** | **69.3** | **75.6** |

Table 9: Results of CricaVPR and the CricaVPR boosted by ImAge (*i.e.*, CricaVPR+ImAge).

| Method | Pitts30k | | | MSLS-val | | | Nordland | | |
|---|---|---|---|---|---|---|---|---|---|
| | R@1 | R@5 | R@10 | R@1 | R@5 | R@10 | R@1 | R@5 | R@10 |
| CricaVPR | **94.9** | 97.3 | **98.2** | 90.0 | 95.4 | 96.4 | 90.7 | 96.3 | 97.6 |
| CricaVPR+ImAge | **94.9** | **97.5** | **98.2** | **92.0** | **97.2** | **97.3** | **94.1** | **97.9** | **98.7** |

## D Comparison to NetVLAD Using Other Transformer Backbones

In the main paper, we conduct experiments using the DINOv2 backbone to validate the effectiveness of our method. Notably, DINOv2 is a foundation model based on the ViT architecture and pre-trained on the large-scale curated LVD-142M dataset. However, our method is also applicable to other transformer models. To this end, we conduct additional experiments using the CLIP [57] model and a ViT model pre-trained only on ImageNet. The results are shown in Table 8. We observe that our proposed method, ImAge, consistently achieves higher R@1 performance across all datasets compared to the explicit aggregation method NetVLAD. However, the performance gains of ImAge are less pronounced (except for ViT-ImAge on MSLS-val) compared to using DINOv2-base-register as the backbone. This observation aligns with the prior study [40], which suggests that foundation models pre-trained on large-scale datasets (significantly larger than ImageNet) are more capable of utilizing additional tokens to capture global information. Moreover, using CLIP as the backbone yields significantly less improvement than DINOv2. Although CLIP is a widely used foundation model (*i.e.*, a vision-language model), its pre-training data and objectives differ considerably from those of the VPR task, making it not a promising choice. This is consistent with the prior work AnyLoc [39], which suggests that CLIP performs significantly worse than DINOv2 in outdoor VPR scenarios.

## E Improving Other VPR Methods with ImAge

Since our ImAge is a general image representation method for VPR, it can not only be implemented based on different transformer backbones, but also can be combined with some other VPR methods to improve their performance. This section uses the CricaVPR [49] method as an example to conduct experiments, and the results are shown in Table 9. It can be seen that our method significantly improves the performance.

## F The GPU Memory Usage and Computational Efficiency in Training

Our method does not add agg tokens before the first block of the transformer backbone, which can significantly reduce GPU memory usage and computational burden. Here, we not only compare our method with adding tokens before the first block, but also use NetVLAD and SALAD as baselines. The results are in Table 10. Our method has significant advantages over adding tokens before the first block in terms of GPU memory usage and training time, and also outperforms NetVLAD and SALAD.

Table 10: The comparison of training GPU memory usage and training time.

| Method | Training GPU Memory (GB) | Training Time/Epoch (min) |
|---|---|---|
| NetVLAD | 17.54 | 9.93 |
| SALAD | 21.81 | 9.98 |
| Adding tokens before 1st block | 34.00 | 15.12 |
| ImAge (Ours) | **16.73** | **9.87** |

## G  The Attention Visualization of Aggregation Tokens

Here we provide the visualization of attention weights of our agg tokens to other patch tokens, as in Fig. 5. This vividly demonstrates that our agg tokens can effectively focus on objects beneficial for VPR (*e.g.*, buildings and vegetation) while ignoring irrelevant or even detrimental elements (*e.g.*, sky and moving vehicles). Additionally, we can observe that: 1) Our method maintains consistent attention on key objects under significant illumination and seasonal changes, indicating high robustness. 2) The attention on critical objects is sparse rather than uniform, suggesting that typically only the most distinctive features need to be considered for VPR. Even for buildings, there is no need to focus on (aggregate) their full area. 3) Some agg tokens focus on both buildings and vegetation, and there are also multiple tokens that focus on buildings. Therefore, there is not a one-to-one correspondence between agg tokens and human-defined object categories.

## H  Additional Qualitative Results and Failure Cases

In this section, we provide additional qualitative results (*i.e.*, visual examples) as a supplement for Fig. 4 in the main paper. As shown in Fig. 6, our ImAge method demonstrates exceptional robustness in retrieving correct database images across various challenging scenarios, including seasonal/viewpoint/lighting variations and occlusions. In contrast to other methods that fail to distinguish critical landmarks or are misled by superficial similarities, the proposed ImAge accurately captures key features (*e.g.*, building textures, positional relationships) to identify right matches.

Moreover, Fig. 7 illustrates some representative failure cases. While our method achieves relatively close retrievals (within 50 meters) in ambiguous natural scenes without distinct landmarks, it occasionally exceeds the predefined threshold (*i.e.*, 25 meters) due to geographic proximity but insufficient visual discriminability. The fourth example, which is the most challenging, involves nighttime images with over-exposure and motion blur, where all methods (including ours) even fail to meet the 50-meter criterion, highlighting persistent challenges in low-quality visual conditions. These results underscore both the advancements of our approach and the remaining difficulties in VPR, which may require increasing the geographical density of image collection for the database to solve. Additionally, for the last two samples, SelaVPR based on local feature re-ranking obtains the correct results, while other methods (including ours) all fail. This points to a possible way to further enhance the robustness of our approach in the future.

## I  More Details about Datasets

The testing datasets used in our experiments, including Pitts30k, Pitts250k, Tokyo24/7, Nordland, SPED, St. Lucia, and SVOX, are organized following the Visual Geo-localization (VG) benchmark [10]. Notably, we use the official version MSLS dataset as in previous work [68, 2, 49, 3, 34]. This version of MSLS-val only consists of 740 query images, which is different from the version in the VG benchmark [10]. In addition, there are also several versions of the Nordland dataset in the VPR task. In our experiments, we use the version in the VG benchmark [10], which employs the summer images as the database and the winter images as queries, each containing 27592 images. Baidu Mall [61] is a well-known indoor dataset for image-based localization. All images are collected at a shopping mall that is over 5000 square meters with many challenging elements, such as transparent windows, reflective materials, repetitive structures, dynamic pedestrians, etc.

Moreover, in the comprehensive comparison (*i.e.*, Table 2 in main paper) with other SOTA methods, we merge Pitts30k-train, MSLS-train, SF-XL, and GSV-Cities for training, following the approach in SelaVPR++ [48]. Specifically, we process datasets other than GSV-Cities to divide places into a finite number of categories, thus facilitating fully supervised training with the multi-similarity loss [67].

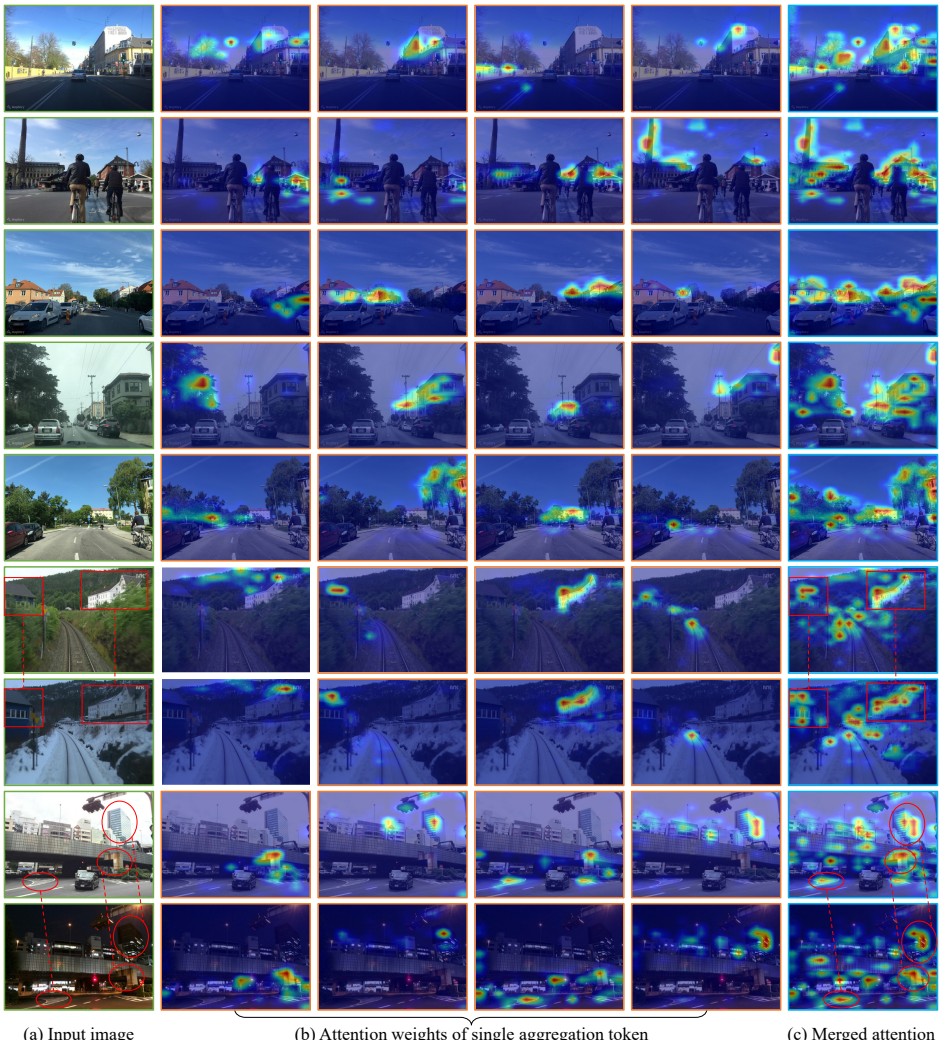

(a) Input image       (b) Attention weights of single aggregation token       (c) Merged attention

Figure 5: The visualization of the attention weights of our agg tokens to patch tokens. The first column (a) represents the input images. The middle 2-5 columns (b) separately display the attention weights of a single agg token to all patch tokens (reshaped to restore spatial position), meaning each image shows the attention of only one agg token. The last column (c) shows the merged attention of all 8 agg tokens. The first five examples (*i.e.*, five rows) show five different places, with buildings, vegetation, and dynamic interference. While different agg tokens attend to distinct regions (or objects) in the images, they consistently focus on stable and discriminative areas (*e.g.*, buildings and vegetation), while largely ignoring variable elements (*e.g.*, cars). The sixth and seventh examples show two images taken at the same place in different seasons. Our agg tokens can consistently focus on buildings (and some discriminative regions where the terrain and railroad tracks change). The last two examples demonstrate that agg tokens can consistently focus on buildings and landmarks even after undergoing severe lighting changes.

## J    More Details about Compared Methods

In the main paper, we compare our method with several other approaches and briefly introduce them. Here, we provide more details about them (for the results in Table 2).

**NetVLAD** [4] and **SFRS** [28] both consist of a VGG16 backbone and a NetVLAD aggregator, and use Pitts30k as the training dataset. The latter employs self-supervised image-to-region similarities to mine hard positive samples for training a more robust model. In our experiments, we use their PyTorch implementations for comparison.

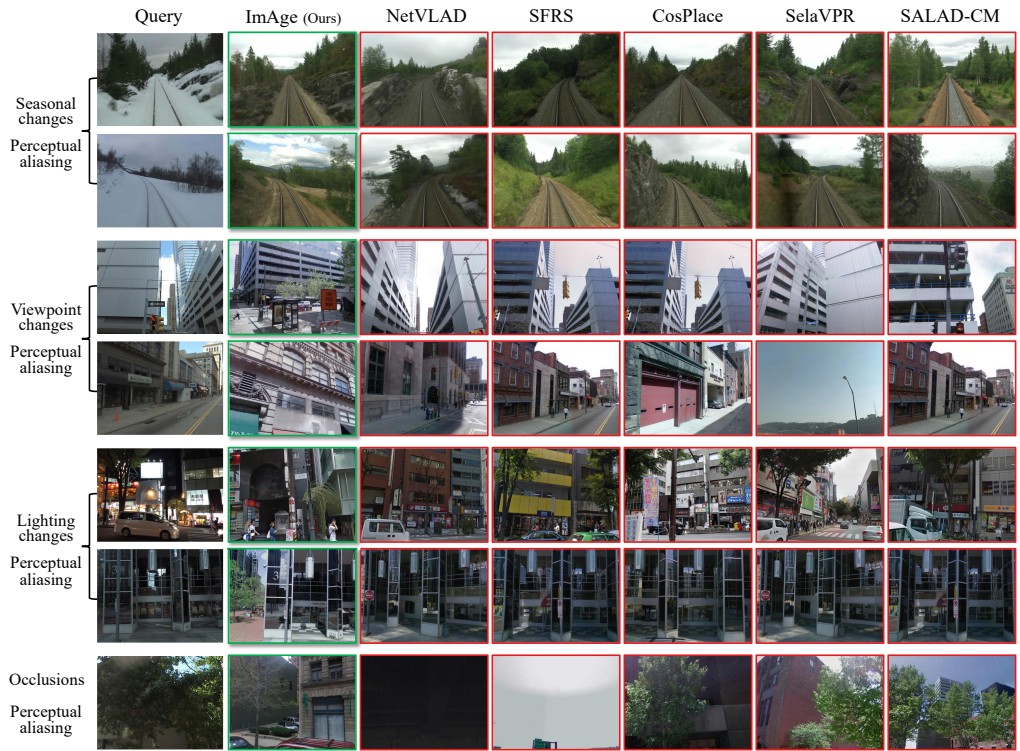

Figure 6: Qualitative results. In four challenging groups of examples (covering seasonal changes, viewpoint variations, lighting changes, occlusions, etc.), our ImAge successfully retrieves the correct database images, while other methods fail. In the first group, most other methods return incorrect images with landscapes absent from the query image (*e.g.*, lakes, cliffs, and hills) or railway tracks of contradicting shapes. The examples in the second group exhibit significant viewpoint variations, where our ImAge consistently gets the right results and demonstrates high robustness. In contrast, other methods return images that appear similar in viewpoint but are actually wrong. Still, they cannot distinguish the critical difference of the landmarks (*e.g.*, the texture of the buildings and their positional relationship). As for the third group, the dim nature of the query image likely interferes with the judgment of the other approaches, resulting in low-luminosity images with different buildings. Dynamic objects like cars in the first example query of this group are also misleading. Nevertheless, our method successfully caught the key features (*e.g.*, the texture of buildings). The final group shows a complex query with severe occlusions by a colossal tree. It is so difficult that all these methods except ours have crashed, returning perceptually similar but wrong images that are also extensively covered (by darkness, brightness, and trees). In summary, our ImAge method demonstrates unparalleled capacity to recognize the truly identical place against various perceptual variations.

**CosPlace** [8] and **EigenPlaces** [11] both frame VPR training as a classification task and use the SF-XL dataset to train their models. For Cosplace, we use the official model based on the VGG16 backbone (with the 512-dim output feature) for testing. For EigenPlaces, we utilize its official implementation and the best configuration based on the ResNet50 backbone to output 2048-dim global descriptors for comparison.

**MixVPR** [2] aggregates the deep features using the multi-layer perceptrons and trains the model with multi-similarity loss [67] on the GSV-Cities [1] dataset. We apply its best-performing configuration (ResNet50 with 4096-dim output features) for comparison.

**CricaVPR** [49], **SuperVLAD** [51], **SALAD** [34], **BoQ** [3], and **EDTformer** [37] all use the foundation model DINOv2-base [55] as the backbone to extract deep features, and train their models on GSV-Cities with the multi-similarity loss. In the comparison experiments, we consistently use their official implementations and the best configurations.

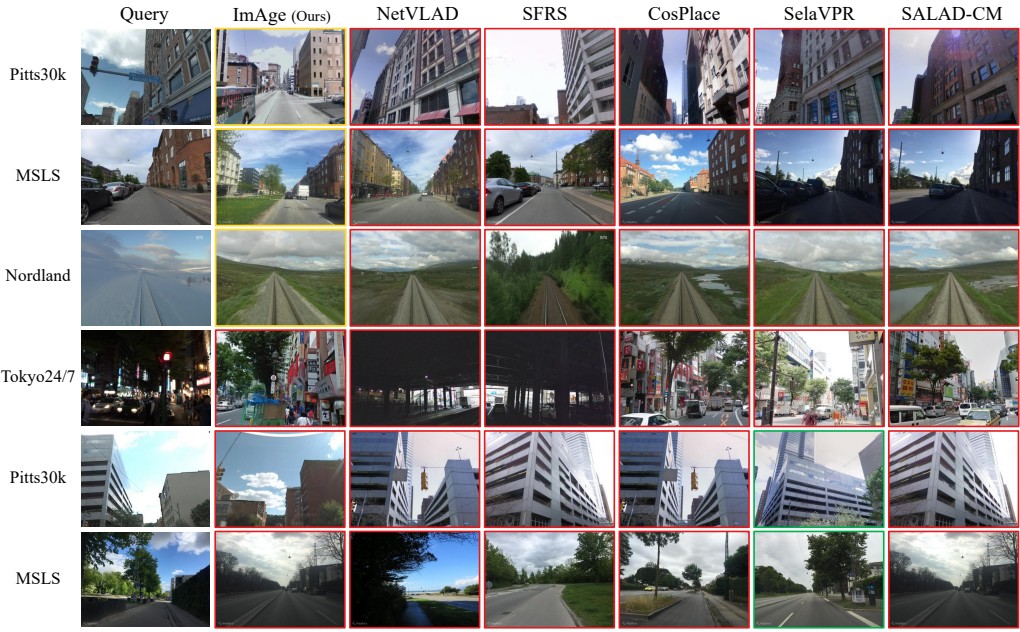

Figure 7: Failure cases. In the first three examples, our method retrieves database images that are geographically close to the query images. However, the distance (radius) between these retrieved images and the query images exceeds the predefined threshold (*i.e.*, 25 meters), although it remains below 50 meters. These cases (partially correct) are labeled in yellow. For the first two cases, ImAge tolerably retrieves results with distances of 33.11 meters and 27.32 meters, while other methods even fail to find an image within 50 meters. In the third example, the images are captured in natural scenes without discriminative landmarks. Nonetheless, ImAge can effectively exclude incorrect answers involving ponds and rivers, while some other methods fail to do so. The distance for this retrieval is a fair 35.70 meters, compared to other methods ranging from 161.90 meters to 4592.90 meters. In the fourth challenging example, all methods, including ours, fail to get an answer within 50 meters. This challenge arises from the complex lighting conditions at night, where over-exposure in bright areas, such as lights, affects the overall texture and the visibility of landmark details. For the last two challenging cases (involving large changes in viewpoint), all methods (including ours) that rely solely on global features for retrieval fail. Notably, SelaVPR, which is based on local features re-ranking, yields the right results. This provides a potential direction for further improving the accuracy of our method. In short, some challenges for current VPR methods remain, despite our method moving a step forward from others.

**SALAD-CM** [33] is an improvement of SALAD. This work analyzes the Geographic Distance Sensitivity of VPR embeddings and proposes a novel mining strategy to address it. Moreover, SALAD-CM first trains the model using both the GSV-Cities and MSLS datasets for better performance. In the comparison experiments, we follow its official implementation.

The rest **TransVPR** [66] and **SelaVPR** [50] are two-stage VPR methods. These works provide two models: one trained for testing on urban datasets (*e.g.*, Pitts30k and Tokyo24/7), and another trained for testing on datasets that may contain suburban and natural scenes (*e.g.*, MSLS and Nordland). We follow the usage in their original paper for comparison experiments.

## K    The Snapshot of MSLS Leaderboard

Fig. 8 is the snapshot of the MSLS place recognition challenge [68] leaderboard at the time of submission, and our ImAge method ranks 1st.

| Results | | | | |
|---|---|---|---|---|
| # | User | Entries | Date of Last Entry | recall@5 ▲ |
| 1 | ImAge4VPR | 1 | 05/11/25 | 0.94 (1) |
| 2 | SelaVPRplusplus | 3 | 01/31/25 | 0.94 (1) |
| 3 | anonymous456 | 9 | 03/02/25 | 0.94 (2) |
| 4 | amaralibey | 1 | 07/07/24 | 0.90 (3) |
| 5 | mapillary_challenge | 11 | 04/17/24 | 0.90 (3) |
| 6 | SKyxuan | 16 | 07/04/24 | 0.90 (4) |
| 7 | anonymous123 | 9 | 07/08/24 | 0.90 (5) |
| 8 | ningzuotao | 16 | 12/20/23 | 0.89 (6) |
| 9 | magnus | 1 | 06/05/24 | 0.89 (6) |
| 10 | razor | 1 | 06/05/24 | 0.89 (7) |
| 11 | izquierdo | 25 | 11/15/23 | 0.89 (8) |
| 12 | anonymous02 | 3 | 09/17/23 | 0.89 (9) |
| 13 | uno | 30 | 06/12/24 | 0.89 (9) |
| 14 | qixi | 6 | 12/19/23 | 0.89 (9) |
| 15 | Pleiades | 1 | 03/27/25 | 0.88 (10) |

Figure 8: The snapshot of MSLS place recognition challenge leaderboard. Our ImAge method (named "ImAge4VPR" for double-blind policy) ranks 1st at the time of submission.

