# OpenReview forum: "Towards Implicit Aggregation: Robust Image Representation for Place Recognition in the Transformer Era"
_NeurIPS.cc/2025/Conference — NeurIPS 2025 poster_

### Official Review · Reviewer_tSze · 2025-06-29

**Clarity:** 3
**Significance:** 3
**Originality:** 3
**Rating:** 5
**Confidence:** 3

**Summary:**

The paper proposes a novel global descriptor for visual place recognition made of the concatenation of "learnable tokens" / registers stacked to the visual tokens of a visual-transformer-network.
The "regular" visual tokens are discarded.

The design of such global descriptor contrasts with existing architectures that follow a 2 step approach: extract visual features with a backbone and aggregate them with an aggregator network.
One issue that can arise then is any domain shift in the backbone feature extract hinders the performance of the aggregation network.
Defining the global descriptor from the backbone network only addresses this issue.

The additional tokens can be stacked to the regular ones at different levels of the transformer network.
The paper motivates and demonstrates through experiment that the optimal results are obtained when the tokens are inserted after the last frozen layer.
The global descriptor is best when the learnable tokens are initialized based on the frozen layer outputs: clustering the frozen layer's feature space and using the centroid to initialize the learnable tokens leads to the best results.

The proposed global descriptor is evaluated following the visual place recognition guidelines (evaluation datasets, metrics, relevant and exhaustive baselines).
The results support the claim of the paper in that the proposed aggregation-free and register-based descriptors are powerful representations for visual place recognition and perform better or on than other transformer-based descriptors.

**Questions:**

- In Eq3, is there an interpretation to the term $Q_z K_a^T V_a$ that also reprensets interactions between the aggregation token / registers and the image embeddings?

**Ethical Concerns:**

["NO or VERY MINOR ethics concerns only"]

**Final Justification:**

Thank you for the rebuttal.
The rebuttal has addressed all the concerns, mainly the confusion in some of the writing and to remove some unjustified claims.
The rating remains the same.

**Limitations:**

Yes, the authors have adequately addressed the limitations and potential negative societal impact of their work (in the supp. material)

### Misc. Comments

Undefined terms:
- L46: "global optimiality" of a global descriptor
- L48: "correction or refinement" of a global desctiptor. Is this supposed to mean "finetuned"?


Unjustified claims:
- L50 " With regard to specific methods like NetVLAD, there exist some problems, such as [...] the loss of position information of the original patch features.". Why is that an issue? If this is related to re-ranking, there are works that enable re-ranking from the global descriptors only (e.g. "Global Features are All You Need for Image Retrieval and Reranking", ICCV23)

- L354: "These centers, obtained via unsupervised clustering of descriptors from randomly sampled training images, tend to capture common visual patterns (e.g., buildings, vegetation, vehicles, pedestrians, etc)."
   - how is it proved that these descriptors capture human level semantics?

References:
- L103: BoW's correct reference is not [4] but

Sivic, and Zisserman. "Video Google: A text retrieval approach to object
matching in videos." Proceedings ninth IEEE international conference on
computer vision. IEEE, 2003.

- L30: [52] is a generic place recognition method so it is not a relevant citations related to the existing challenges in VPR

**Quality:**

3

**Strengths And Weaknesses:**

S1. Novel and interesting design of the global descriptor that demonstrate how visual transofrmers can learn global descriptors through register tokens.
The design is well motivated and the experiments demonstrate that it is suitable for place recognition.

S2. Exhaustive experimental evaluation that support the claim of the paper.

S3. Relevant related work.

S4. Well written paper.


W1. The paper claims to address the "limitations of the previous backbone-plus-aggregator" but the justifications are not sound so this claim does not hold.
It is best to simply state that the proposed descriptors is novel because it learns global tokens within the transformer architecture, which is a good contribution on its own.

L81: "This aggregator-free VPR paradigm addresses the limitations of the previous backbone-plus-aggregator paradigm by:"
- a) "unifying feature extraction and aggregation toward global optimization, and"
    - Aggregation-based method enable global optimization by making the aggregation differentiable (e.g. NetVLAD, BoQ, ...) so this sentence is incorrect.
- b) "progressively implementing aggregation in cascaded transformer blocks, thus achieving the correction and refinement of image representations."
    - "correction and refinement of image representations" is undefined. If this means improving the feature representations with each layers, all networks (CNN, VIT ...) do so

L91: "This new paradigm effectively and efficiently overcomes the limitations of the previous works".
  - Same comments as before: the design of previous works, i.e., aggregation-based descriptors, is not a limitation per se

---

> ### Author Rebuttal · Authors · 2025-07-30
>
> Thanks for your positive recommendation and detailed suggestions. The following are Responses to the Questions (Q), Weaknesses (W), and Misc. Comments (M).
>
> **Q. about the term $Q_z K_a^T V_a$**
>
> We truly appreciate your attention to this detail. Indeed, this term also represents interactions between agg tokens and the image embeddings, but it is the influence of the former on the latter (unlike $Q_a K_z^T V_z$). Since we leverage the inherent self-attention mechanism within the backbone to jointly process all tokens, bidirectional interactions between agg tokens and image embeddings are unavoidable. When our agg tokens aggregate useful global information from the image embeddings, the image embeddings can also receive information from the agg tokens. This behavior is analogous to the interaction between the [CLS] token and patch tokens in the standard ViT architecture.
>
> We did not explain this item in the method section to avoid confusing the readers. However, in our ablation studies, we explored the impact of setting different numbers of agg tokens.
> The results show that agg tokens have no negative impact on image embeddings as long as the number of tokens is not too large. When the number of agg tokens becomes excessively large (e.g., 64), a noticeable decline is observed. This indicates that an excessive number of agg tokens can interfere with the contextual information of patch tokens, which is not recommended. It should be noted that an excessively large number of agg tokens will cause serious overhead in model inference, and the high-dimensional descriptors generated by these tokens will also cause serious overhead in retrieval. So, this setting does not meet the actual needs. That is, it is not necessary to view it as a negative impact. Several agg tokens have already been able to effectively aggregate the useful information of patch tokens.
>
> **W1. The limitations of the previous method**
>
> Thank you for recognizing the novelty and contribution of our approach. We will follow your suggestion and directly state the novel perspective of our proposed method, removing some claims of the limitations of previous methods.
>
> -	L81: a) “toward global optimization”. --> We agree with you that this sentence is incorrect. We should claim “unifying feature extraction and aggregation into a more cohesive framework”, rather than “toward global optimization”.
>
> -	L81: b) “correction and refinement of image representations”. --> Sorry to confuse you. The image representations here refer to the global image feature (i.e., a representation vector).
> Although other methods improve the feature representations with each layer, what is improved is the patch descriptors (i.e., feature maps in CNN, patch tokens in ViT). The global image representation is formed after one-shot processing by the aggregation layer and is immediately output (without opportunity for correction and refinement). Our method, however, inserts agg tokens in the backbone's L-th encoder layer. These agg tokens serve as global image representations, and they are subsequently corrected and refined in subsequent encoder layers (synchronously with the refinement process of patch tokens), rather than being formed all at once. This is indeed different from the process of forming global representations in the backbone+aggregator approach. We will explain this in detail in the final paper to avoid confusion.
>
> -	L91: ”… overcomes the limitations of the previous works”. --> We agree with you. We will delete “overcomes the limitations” and will revise this sentence as “Our method provides a novel perspective different from the previous paradigm, effectively and efficiently achieving better performance in the transformer era.”
>
> **M1. Undefined terms**
>
> -	"global optimality".  --> Please see the response for W1 (L81 a)). We will delete this statement according to your suggestion.
> -	"correction or refinement".  --> Please see the response for W1 (L81 b)). "fine-tuned" is also OK, but we are concerned that it might easily be confused with the training of models.
>
> **M2. Unjustified claims**
>
> -	“The issue with NetVLAD” --> We acknowledge that some improvement efforts in the last two years have made the issue of the loss of position information less of a concern. However, the loss of position information objectively exists in NetVLAD/GeM, but our method does not have this issue. This is indeed a minor issue and is not sufficient to serve as the highlight/motivation of our approach. We will remove these statements from the main paper.
> -	“How is it proved that these descriptors capture human level semantics?” --> Sorry to confuse you. Although we use unsupervised clustering of patch descriptors during initialization, the transformer backbone used to extract these patch descriptors is pre-trained on large-scale datasets, which can indeed provide high-level semantics [56]. One way to demonstrate this is to visualize the attention weights of agg tokens to patch tokens, observing which patches are most correlated with the agg token. The supplementary materials we submitted (a PDF in the zip file) show that agg tokens from the trained model only focus on objects that are helpful for VPR, which provide vivid and interesting qualitative results. Before training (after initialization), the agg tokens focus on not only buildings and vegetation, but also vehicles and pedestrians, i.e., common visual patterns. However, it should be noted that it does not mean that an agg token exactly corresponds to a human semantic object (e.g., the building or vegetation). From attention visualization, we can see that some tokens contain both buildings and vegetation, and there are also multiple tokens that focus on buildings. We will emphasize this point in the final paper to avoid misleading.
>
> **M3. References**
>
> We sincerely apologize for citing secondary sources rather than the original work in the two instances you pointed out. For the first instance, we will cite the BoW work you mentioned. For the second instance, we will cite the work [47].
>
> **Thanks again for your valuable comments and suggestions. Please let us know if you have any further concerns. We'd be very glad to address them.**
>
> **`Reference`**
>
> [47] Stephanie Lowry, et al. Visual Place Recognition: A Survey. IEEE Transactions on Robotics, 2015.
>
> [56] Maxime Oquab, et al. DINOv2: Learning Robust Visual Features without Supervision. Transactions on Machine Learning Research Journal, 2024.

---

> ### Comment · Reviewer_tSze · 2025-08-05
> **Rebuttal answers the few questions and addresses the confusing writing**
>
> Thank you for the rebuttal. The rebuttal has addressed all the concerns, mainly the confusion in some of the writing and to remove some unjustified claims. The rating remains the same i.e. accept

---

> > ### Author Response · Authors · 2025-08-06
> >
> > We are pleased that our response has addressed all your concerns. Thanks a lot for maintaining the "Accept" rating. We will incorporate your valuable suggestions into our revised paper.

---

### Official Review · Reviewer_zpn8 · 2025-06-30

**Clarity:** 3
**Significance:** 1
**Originality:** 1
**Rating:** 2
**Confidence:** 5

**Summary:**

This paper proposes ImAge, a framework for visual place recognition that eliminates the need for explicit aggregators. Instead, it leverages the self-attention mechanism of a transformer backbone by inserting learnable aggregation tokens at a specific depth.

**Questions:**

Please see the above weaknesses.

**Ethical Concerns:**

["NO or VERY MINOR ethics concerns only"]

**Final Justification:**

I indeed appreciate the authors’ further responses. However, it does not impress me substantively. I still insist that many papers have proposed similar CLS tokens and insert strategies in Transformer. An engineering modification for VPR task is insufficient to meet the acceptance standards of NeurIPS.

**Limitations:**

The discussion of the limitations in this paper is insufficient and lacks transparency. The authors claim that “ImAge is broadly applicable to a wide range of image retrieval tasks,” but this assertion appears somewhat inverted in logic. In fact, prompt tuning methods have already been widely applied to mainstream vision tasks such as image retrieval, classification, and object detection. The proposed ImAge method essentially transfers the prompt tuning strategy to the relatively niche task of visual place recognition (VPR). However, the authors do not provide a systematic comparison between ImAge and existing prompt tuning approaches in other vision domains, limiting their analysis and validation to the VPR setting alone.

**Paper Formatting Concerns:**

No formatting issues in this paper

**Quality:**

1

**Strengths And Weaknesses:**

Strengths：

The paper is well-written, with clearly articulated ideas that are easy to follow.

Weaknesses：

[1] The paper tends to rely heavily on storytelling, while the proposed method is essentially a form of prompt tuning. Compared to existing classical prompt tuning approaches (e.g., VPT [R1]), it lacks substantive innovation, offering only engineering-level adjustments to the number, position, and initialization of prompt tokens. Applying this method to visual place recognition (VPR), a classification task, is fairly intuitive. However, the authors only compare their method against other VPR-specific approaches, deliberately avoiding comparisons with established prompt tuning methods in the broader computer vision community.

[2] The comparative experiments are not sufficiently rigorous. As shown in Table 6, the model's performance is highly sensitive to the number of aggregation tokens. Nevertheless, the ablation studies presented in Tables 4, 5, and 7 are only conducted under the setting where the number of agg tokens is 8. Therefore, the conclusions drawn from these ablations may lack generality and robustness.

[3] The proposed initialization strategy for cluster centers is inspired by NetVLAD, but its rationale is not well justified. In fact, the mechanism for computing global representations in the two methods differs fundamentally: NetVLAD constructs global descriptors using residual statistics between patch features and cluster centers, whereas ImAge directly flattens the cluster centers to form the global representation. This conceptual gap is not clearly explained, which undermines the credibility of the design.

[4] Moreover, the proposed method is highly sensitive to multiple hyperparameter choices, including the insertion strategy, initialization, and number of aggregation tokens. Due to the lack of theoretical justification, determining these key hyperparameters in new practical scenarios remains a challenge, which may significantly affect the model’s stability and generalization.

Reference：
Jia, Menglin, et al. "Visual prompt tuning." European conference on computer vision. Cham: Springer Nature Switzerland, 2022.

---

> ### Author Rebuttal · Authors · 2025-07-30
>
> Thanks for your comments to help us improve the paper. The following are Responses to the Questions/Weaknesses.
>
> **W1. About the novelty of ImAge**
>
> We sincerely appreciate your suggestion of conducting an experimental comparison between ImAge and prompt tuning (the results are in the table below). Regarding the novelty concerns, you might have some misunderstandings about our method. Before explaining the differences between our method and prompt tuning, we need to point out that VPR is an **image retrieval task**, not a classification task as you said. This might have led to some of your confusion about the sensitivity of our method (please see response to W2). To clarify the differences between our method and prompt tuning, we provide the following detailed explanations.
>
> -	We acknowledge VPT is an excellent parameter-efficient fine-tuning (PEFT) method applicable to many downstream tasks. However, it does not perform well in the VPR task as the prior work [1] demonstrated. The PEFT method, including prompt tuning, has been widely studied in the VPR task for as early as 1-2 years ago [2][1]. If our method is prompt tuning, similar performance would have already been achieved by prior work. However, the fact is that our method significantly outperforms previous methods and ranks first on the leaderboard of MSLS place recognition challenge.
> -	In terms of methodology, our approach differs from prompt tuning. Like several recent SOTA methods, our method adapts a pre-trained model to the VPR task by fine-tuning the last several blocks of the transformer backbone, rather than using prompts. It differs from prompt tuning where "only the task-specific prompts are being updated during fine-tuning, while the transformer backbone is kept frozen" [5]. The added agg token in our method is used to aggregate the key information in the patch tokens, rather than as the prompt.
> -	In fact, apart from prompt tuning, there are other works that add additional tokens to the pre-trained transformer for purposes other than fine-tuning the model. We have discussed this in detail in the second part of the Related Work. Our approach is more relevant to the other two types of methods than prompt tuning. We also discussed the differences between our approach and other methods in  Related Work, Section 3.2 (Lines 187-190), and Appendix C (especially in this part). In fact, our agg tokens are “output-oriented tokens” rather than “prompt tokens”. Meanwhile, our work is inspired by the use of memory/register tokens in DINOv2-register [3] to capture global information.
> -	Besides, we also conduct fair comparative experiments to demonstrate that our method is different from prompt tuning (VPT). For applying VPT to the VPR task, an aggregation layer is necessary. We use the NetVLAD aggregation layer to obtain the final image descriptor. We trained two model versions of VPT, i.e., VPT (shallow) and VPT (deep). All models are based on the DINOv2-register backbone and trained on GSV-Cities, as in Table 3 of our paper. The results in the table below show that the performance of VPT is limited (basically consistent with [1]). The three methods in the last three rows (including our ImAge) fine-tune the DINOv2-register backbone by fine-tuning the last four encoder blocks (widely used by SOTA methods in VPR), achieving significantly better performance than prompt tuning (VPT).
> | Method | Pitts30k | MSLS-val | Tokyo24/7 | Nordland |
> |---|---|---|---|---|
> | VPT (shallow) + NetVLAD | 83.5/92.5/94.8 | 53.8/67.2/70.9 | 61.6/81.0/84.4 | 23.6/38.0/45.2   |
> | VPT (deep) + NetVLAD  | 90.8/95.7/97.0 | 84.7/93.1/95.0 | 92.4/97.8/98.1 | 52.5/67.8/74.1  |
> | NetVLAD | 92.8/96.6/97.8 | 91.8/96.5/96.6 | 95.6/98.1/98.7 | 90.5/96.5/97.8 |
> | SALAD | 92.5/96.6/97.5 | 92.6/96.6/97.0 | 95.6/97.5/99.0 | 86.5/93.6/95.7|
> | ImAge | 94.0/97.2/98.0 |93.0/97.0/97.2 | 96.2/98.1/98.4 | 93.2/97.6/98.6 |
>
> -	In short, the prompt tuning methods use the prompts for fine-tuning the pre-trained large model, rather than as the final representation for downstream tasks. In contrast, our ImAge directly uses the agg tokens as the final global representation, which presents an opposite design perspective. Besides, our proposed insertion and initialization strategies are not the ready-made method used in the prompt tuning research or simple engineering tweaks, but thoughtful designs to enable the added agg tokens to aggregate useful information from the patch tokens. Here, we also use the output tokens corresponding to the prompt tokens in VPT as final image descriptors, and the results are shown in the following table. There is still a significant gap compared to our method.
> | Method | Pitts30k | MSLS-val | Tokyo24/7 | Nordland |
> |---|---|---|---|---|
> | prompt tokens in VPT | 80.5/92.2/94.8 | 62.4/74.2/79.7 | 55.9/73.0/81.3 | 13.0/21.6/26.6 |
> | ImAge | 94.0/97.2/98.0 |93.0/97.0/97.2 | 96.2/98.1/98.4 | 93.2/97.6/98.6 |
>
> **W2. About the ablation studies**
>
> We appreciate your concern about the robustness of our method. In standard ablation studies, the common practice is to fix all variables except the one being studied, in order to avoid the influence of other factors. In the prompt tuning research ([4], [5]) you mentioned, they were also fixing the prompt length when exploring the influence of other factors. As for your observation that "the model's performance is highly sensitive to the number of aggregation tokens" in Table 6, we would like to clarify that VPR is a retrieval task, and the performance of the method is directly influenced by the dimension of the image descriptors (the global feature vector). This holds true in all other VPR studies. The different numbers of tokens in our method directly form different dimensions of image descriptors, and this is reasonable in terms of its impact on performance. This is most pronounced on the Nordland dataset (very difficult), which is known to benefit from high-dimensional global descriptors (to provide detailed information). So, as you can see in Table 4, our method does not perform well using low-dimensional descriptors (i.e., only 1 agg token). In fact, this is also true for other methods. Compared with other low-dimensional descriptors, the results of our method on Nordland are actually better, e.g., the baseline (CLS token) in this table is 11.3% lower than our method on R@1. Moreover, on more commonly used benchmarks like Pitts30k and MSLS-val, our ImAge shows much lower sensitivity to the number of agg tokens. In addition, high descriptor dimensions will cause a significant storage burden and computational time consumption for retrieval tasks. Taking all aspects into consideration, we finally chose 8 agg tokens as the default setting. This results in a 6144-dimensional descriptor, which is superior to recent works (SALAD is 8448-dim and BoQ is 12288-dim).
>
> **W3. About the initialization strategy**
>
> Sorry to confuse you. We appreciate your concern for the rationale behind our initialization strategy. We also agree that NetVLAD constructs global descriptors using residual statistics between patch features and cluster centers, which is different from that our ImAge directly concatenates agg tokens as global descriptors. However, we merely drew inspiration from the initialization of NetVLAD, and this initialization is not unique to NetVLAD. It is actually the widely used k-means clustering algorithm. Since these cluster centers represent common visual categories and provide more visual prior, we use the cluster centers to initialize our learnable agg tokens. Such initialization can provide guidance for agg tokens to learn meaningful global information better. This is also partially discussed in our method section and ablation studies section.
>
> **W4. Sensibility to hyperparameter choices**
>
> As our response to W2, the performance of all methods in retrieval tasks varies with the descriptor dimensionality (corresponding to the number of agg tokens in our method). Regarding the insertion strategy and initialization method, it is precisely because other methods are not robust enough in the VPR task that we have proposed our insertion strategy and initialization method. **It should be noted that in Tables 4 and 5, only the best one is our method**. The other results correspond to other methods rather than different Settings of our method. This might be the reason for your misunderstanding. We have explained the reason for the specific design in the Method section. Ablation experiments demonstrate that our method is highly robust to various VPR scenarios, e.g., the Pitts30k (urban scene), MSLS (suburban and urban scene), and Nordland (natural scene) datasets. In addition, we are also happy to apply our method to more scenarios to verify its robustness.
>
> **Limitations**
>
> As our response to W1, our method is different from prompt tuning. If you agree that our method is a new implicit aggregation approach to obtain global features in retrieval tasks, we believe you will also recognize that it is likely to be equally effective in other retrieval tasks. As far as we know, this implicit aggregation method that does not require an additional aggregator has not yet been used for other retrieval tasks.
>
> **Thanks again for your valuable comments and suggestions. Please let us know if you have any further concerns. We'd be very glad to address them.**
>
> **`References`**
>
> [1] Wu, Rouwan, et al. Investigating the Different Method of Parameter-Efficient Fine-Tuning for Pre-Trained Vision Models on Visual Place Recognition. Available at SSRN 5233632.
>
> [2] Lu, Feng, et al. Towards seamless adaptation of pre-trained models for visual place recognition. ICLR 2024.
>
> [3] Darcet, Timothee, et al. Vision Transformers Need Registers. ICLR 2024.
>
> [4] Lester, Brian, et al. The Power of Scale for Parameter-Efficient Prompt Tuning. EMNLP 2021.
>
> [5] Jia, Menglin, et al. Visual prompt tuning. ECCV 2022.

---

> > ### Author Response · Authors · 2025-08-06
> >
> > Dear Reviewer zpn8,
> >
> > We sincerely appreciate your valuable time and effort to review our paper. We have carefully provided detailed responses to your comments, including the difference between our method and prompt tuning, as well as the experimental comparisons demonstrating that our method has a significant advantage over prompt tuning in the VPR task. We hope that our responses have addressed all your concerns. As the reviewer-author discussion deadline is approaching, please let us know if you have any other concerns. We will make every effort to address them as soon as possible. Thank you very much.
> >
> > Best Regards,
> >
> > Authors

---

> > > ### Comment · Reviewer_zpn8 · 2025-08-06
> > >
> > > Thank you for the detailed responses provided. However, I still have a few remaining concerns.
> > >
> > > **Response to "W1. About the novelty of ImAge":**
> > > DINOV2-register conducts both visualization and quantitative experiments to analyze what information the register tokens have learned. However, this paper lacks any such insightful analysis to explain the role and mechanism of the so-called aggregation tokens. As a result, the paper falls short in providing a theoretical understanding of the proposed method. Overall, the performance gains appear to stem largely from engineering choices—such as tuning the number of aggregation tokens and the number of trainable layers, rather than from algorithmic innovation.
> > >
> > > **Response to "W2. About the ablation studies":**
> > > The authors state: "This is most pronounced on the Nordland dataset (very difficult), which is known to benefit from high-dimensional global descriptors (to provide detailed information)." However, as shown in Table 6, increasing the number of aggregation tokens actually leads to a significant performance drop on the Nordland dataset, contradicting the authors’ claim. Moreover, if the authors argue that “the performance of the method is directly influenced by the dimension of the image descriptors,” then the descriptor dimension should have been kept constant to isolate the effect of the aggregation strategy. In fact, performance degrades substantially when either increasing or decreasing the number of aggregation tokens, suggesting that descriptor dimensionality is not the primary factor driving performance. The current experimental design does not adequately support the authors’ conclusions.
> > >
> > > **Response to "W3. About the initialization strategy":**
> > > The response to W3 does not directly explain why ImAge adopts the same initialization strategy as NetVLAD, despite the fact that their aggregation mechanisms are not computationally equivalent.

---

> > > > ### Author Response · Authors · 2025-08-08
> > > >
> > > > Dear Reviewer zpn8,
> > > >
> > > > We respectfully inquire whether our latest response has addressed your remaining concerns. We sincerely hope that you can take the time to review the visualization result in our supplementary material (a zip file, not the appendix), as it would be very helpful in addressing your first concern. Since the reviewer-author discussion deadline is approaching, please let us know if you have any other concerns. Thanks again for your time and effort to review our work.
> > > >
> > > > Best Regards,
> > > >
> > > > Authors

---

> > > > > ### Author Response · Authors · 2025-08-09
> > > > >
> > > > > Dear Reviewer zpn8,
> > > > >
> > > > > We are still waiting for your feedback on our latest response, which is very important to us. We understand that you may be busy with other things. Whenever you provide your new feedback, we will respond as soon as possible. If most of your concerns have already been addressed, may we kindly ask you to reconsider the value of this work and possibly raise the score? In particular, we have clarified the difference between this work and prompt tuning, as well as demonstrated the role and mechanism of agg tokens by visualization results in the supplementary material file. Thanks again for your time and effort to review our work.
> > > > >
> > > > > Best Regards,
> > > > >
> > > > > Authors

---

> ### Author Response · Authors · 2025-08-06
>
> Many thanks for your detailed feedback. We understand your concerns, and the following responses may help address them.
>
> **For R1:**
>
> We agree that it is necessary to provide analysis/evidence to explain the role and mechanism of the aggregation tokens. The supplementary material we submitted (a PDF in the zip file) has shown the attention visualization of agg tokens to image patches (patch tokens), which provides vivid and interesting qualitative results to support our claim. While different agg tokens attend to distinct regions (or objects) in the images, they consistently focus on the areas that are important for VPR (e.g., buildings and vegetation), while ignoring variable and useless elements (e.g., cars and sky).
>
> In the main paper (Equation 3 and Lines 180-184), we provide a theoretical introduction about that agg tokens can learn and capture the global contextual information within the patch tokens by agg-patch attention (i.e., the term $ Q_aK_z^{\top}V_z$ in Equation 3). The visualization in supplementary material vividly demonstrates that the patch tokens representing the objects highly related to VPR will be assigned the significant attention scores of agg tokens. That is to say, agg tokens aim to capture the key information from patch tokens related to VPR.
>
> The visualization in supplementary material addressed the concerns of Reviewer tSze and Reviewer WgNP, and was considered by Reviewer VZG8 as convincing. We believe this is also useful evidence to address your concerns. We will place it in the paper (rather than as supplementary material) to better demonstrate our method.
>
> **For R2:**
>
> Thank you for your feedback. Since we leverage the inherent self-attention mechanism within the backbone to jointly process all tokens, bidirectional interactions between agg tokens and patch tokens are unavoidable. When our agg tokens learn and aggregate useful global information from the patch tokens by agg-patch attention (i.e., $Q_aK_z^TV_z$), similarly, the patch tokens can also receive information from the agg tokens by patch-agg attention (i.e., $Q_zK_a^TV_a$), as shown in Equation (3) in our paper. When the number of agg tokens becomes excessively large (e.g., 64), a performance decline can be observed. This is because an excessive number of agg tokens can interfere with the contextual information of patch tokens, which in turn affects the agg tokens in aggregating the information related to VPR.
>
> However, it is not necessary to view it as a negative impact. It should be noted that an excessively large number of agg tokens does not meet the actual needs, as they will cause serious overhead in model inference, and the high-dimensional descriptors generated by these tokens will also cause serious overhead in retrieval. On the contrary,  Several agg tokens have already been able to effectively aggregate the useful information of patch tokens. Meanwhile, it will produce descriptors of several hundred to several thousand dimensions, which is in line with practical needs and exactly at the same level as other SOTA works.
>
> In short, a large number of agg tokens are unnecessary, and we don't need to use this setting at all. The experiment conducted in this setting is primarily to illustrate the issue of bidirectional interactions, as described in Lines 376-379. Of course, we can also eliminate this issue by adding an attention mask. We did not do so mainly to maintain the original structure of the backbone and facilitate scalability to other models.
>
>
> **For R3:**
>
> We think the key to this issue is that the initialization strategy is actually the widely-used k-means clustering algorithm, and is not unique to residual-based aggregation methods such as NetVLAD. For example, the prior work SuperVLAD [52] removes the cluster centers and directly aggregates local features (instead of residuals), which also employs such initialization in its code implementation to achieve better performance. **The common point of these methods** is that they all set multiple categories/clusters (note that different agg tokens in our method are responsible for aggregating different categories that are helpful for VPR). Therefore, they can all be based on unsupervised clustering to provide a common visual prior of categories, thereby better guiding aggregation in subsequent training.
>
> Thanks again for your feedback. To ensure the Rebuttal fully addresses all your concerns, we would be grateful if you could share any additional concerns you might have.
>
> `Reference`
>
> [52] Feng Lu, et al. Supervlad: Compact and robust image descriptors for visual place recognition. NeurIPS 2024.

---

### Official Review · Reviewer_VZG8 · 2025-07-01

**Clarity:** 4
**Significance:** 3
**Originality:** 2
**Rating:** 5
**Confidence:** 5

**Summary:**

The paper introduces ImAge, an innovative implicit aggregation paradigm for visual place recognition (VPR) that leverages the self-attention mechanism of transformers to eliminate the need for explicit aggregators.
This approach addresses critical limitations of traditional "backbone+aggregator" architectures, such as suboptimal two-stage optimization and loss of positional information.

The core innovation lies in:
1. Introducing learnable aggregation tokens that interact with patch tokens via self-attention within transformer blocks.
2. Proposing an optimal token insertion strategy and initialization method to enhance representation robustness.
3. Demonstrating state-of-the-art (SOTA) performance across diverse VPR benchmarks with higher efficiency.

**Questions:**

1. Plz provide the memory and computational efficiency comparison results.
2. Plz provide more ablation studies on ViT-based VPR methods to show the additional performance gains and show more significance of this work in VPR field.

**Ethical Concerns:**

["NO or VERY MINOR ethics concerns only"]

**Final Justification:**

The comments from the authors resolve my concerns about W1 and W2. I will finally give "accept".

**Limitations:**

yes

**Paper Formatting Concerns:**

/

**Quality:**

3

**Strengths And Weaknesses:**

Strengths:
1. Paper writing is good and clear to show the motivation and comparison.
2. Thorough ablation studies and fair comparisons validate the method's robustness and efficiency. The approach ranks 1st on the MSLS challenge leaderboard (Fig. 7), demonstrating real-world applicability.
3. The visualization of the attention weights of agg tokens to patch tokens shows the convincing training proces.

Weakness:
1. No GPU memory and computational efficiency comparison has been shown in quantitative experiments while the authors repeatedly stressed “leading to significant GPU memory and computational overhead”.
2. Although this paper has shown comparison of different aggregators based on standard transformer-backbones ViT and DINO-V2，I am curious about the results of transformer-based VPR methods with agg tokens. For example, R2Former.
3. Such agg tokens are not novel enough for transformer/VLM era. Of course, it's kind of novel for VPR.

---

> ### Author Rebuttal · Authors · 2025-07-29
>
> Thanks for your positive recommendation and valuable suggestions. The following are Responses to the Questions/Weaknesses.
>
> **Q1/W1. GPU memory and computational efficiency**
>
> When introducing our proposed insertion strategy of agg tokens, we mentioned that "if agg tokens are added at the beginning,…, leading to significant GPU memory and computational overhead." This primarily refers to the advantages of our method during training. We highly agree with your suggestion. Here, we not only compared this method with inserting tokens at the beginning, but also used NetVLAD and SALAD as baselines. The results are shown in the table below. Our method has significant advantages over inserting tokens at the beginning in terms of GPU memory usage and training time, and also outperforms NetVLAD and SALAD.
>
> | Method | Training GPU Memory (GB) | Training Time/Epoch (min) |
> |----|---|---|
> | NetVLAD | 17.54 | 9.93 |
> | SALAD | 21.81 | 9.98 |
> | Insert at the beginning | 34.00 | 15.12 |
> | ImAge (Ours) | 16.73 | 9.87 |
>
> It is also worth mentioning that we have demonstrated the advantages of our method in inference (inference time and descriptor dimensionality) in the paper (Figure 1 and Table 3). All these results show that our method outperforms previous methods in many aspects.
>
> **Q2/W2. transformer-based VPR methods with agg tokens**
>
> Thank you for giving this suggestion to improve our paper. In fact, our method can be regarded as a general aggregation method for the transformer backbone. In addition to the experiments in the main paper, Appendix D (Table 7) also shows the results of combining the proposed ImAge method with ViT and CLIP. The R2Former you mentioned uses ViT as the backbone, so it is feasible for our method to improve the performance of its first stage (global retrieval). However, R2Former is a two-stage method with re-ranking using local features. Its training process is elaborate, and the model is trained on 8 Tesla-V100 GPUs, which makes it difficult to combine R2Former with our method during the limited Rebuttal period. Considering that R2Former is an excellent VPR method and ranked first in the leaderboard of MSLS Place Recognition Challenge when proposed (same as our method), and the re-ranking in it can significantly improve the VPR performance, we will discuss the potential of combining it with ImAge in our final paper and try to combine it in future experiments.
>
> To show that our method can replace the aggregation method in other transformer-based VPR methods to achieve better performance, we replace the spatial pyramid GeM pooling feature in the CricaVPR work with our ImAge feature (denoted as CricaVPR+ImAge). The results are shown in the table below. It can be seen that our method significantly improves the performance.
>
> | Method | Tokyo24/7 | | MSLS-val | | Pitts30k | | Nordland |
> |----|---|---|---|---|---|---|---|
> | CricavPR | 93.0/97.5/98.1 | | 90.0/95.4/96.4 | | 94.9/97.3/98.2 | | 90.7/96.3/97.6 |
> | CricavPR+ImAge | 96.5/98.1/98.7 | | 92.0/97.2/97.3 | | 94.9/97.5/98.2 | | 94.1/97.9/98.7 |
>
> In short, our ImAge is a general image representation method for the transformer-based VPR. In addition, when using the foundation model DINOv2 trained on a large-scale curated dataset, the performance improvement of our ImAge over other methods (such as NetVLAD) is more pronounced than when using ViT pre-trained on ImageNet. This suggests that in the future, as the performance of the foundation model improves (with more training data), the advantages of our ImAge may become more apparent.
>
> **W3. Novel for VPR**
>
> Thanks for your recognition that our method is novel in VPR. We acknowledge that there have been some works in other fields previously on adding tokens to the transformer, as we discussed in Related Work. Our work provides a new perspective on the method of adding tokens: achieving implicit aggregation of patch tokens. As far as we know, this work is the first to do so not only in VPR but also in visual retrieval. Furthermore, our method achieves the SOTA performance in a concise way.
>
> By the way, we are so happy to see that you appreciate our SOTA performance, real-world applicability, as well as the interesting attention visualizations of agg token (in Supplementary Material).
>
> **Thanks again for your valuable comments and suggestions. Please let us know if you have any further concerns. We'd be very glad to address them.**

---

### Official Review · Reviewer_WgNP · 2025-07-03

**Clarity:** 3
**Significance:** 3
**Originality:** 3
**Rating:** 5
**Confidence:** 3

**Summary:**

This paper proposes a new approach to adapt vision transformers for visual place recognition. Specifically, the novelty focuses on the pooling/aggregation mechanism.
Rather than using an additional separate pooling mechanism on top of the backbone, this work proposes to use additional "aggregation" tokens that use the same self-attention mechanism already in place within the transformer. The output embeddings of these tokens can then be used as image descriptors for place recognition.
This approach outperforms previous visual place recognition approaches.

**Questions:**

1. Could the authors please address my concerns on misleading statements in the paper detailed in my comment on quality #4? These include how the solution tackles claimed limitations of two-stage approaches, clarifications on the description of the method being “aggregator-free” and having “0 aggregator parameters”, experimental details in the caption of table 2, and the conclusions drawn from figures 4, 5 and 6.
2. Can the authors please clarify how the clustering for initialisation works? Specifically, which features are extracted and used for clustering? Is this a limitation for transferring to new settings?
3. Could the authors please discuss potential limitations from moving away from a two-stage approach? I would expect that this would lead to more challenges if one wants to keep the backbone frozen (e.g., to get good generalisation in new domains, as discussed in [40], or if the backbone is extremely expensive to fine-tune).
4. Can the authors please expand more on the statement from line 244-245: "each added agg token representing an object category". It is unclear to me why we would be able to make that assumption, and why tokens would end up with separated categories to represent.
5.  Can the authors please clarify why the proposed aggregation tokens are not classified as "output-oriented tokens"? They seem to follow the same purpose as class tokens, just with different design choices on location and initialisation.

**Ethical Concerns:**

["NO or VERY MINOR ethics concerns only"]

**Final Justification:**

The rebuttal has addressed my issues with presentation and clarity.

While relying on existing mechanisms, I believe the authors validate the specific design of their approach, and demonstrate strong results that might influence future research on visual place recognition.
As such, I find this work to be valuable and would recommend that it be accepted.

**Limitations:**

The authors discuss some limitations in the appendix, but not in the main paper. Moreover, some additional limitations would be interesting to discuss (see my question 3).

**Paper Formatting Concerns:**

I have no major formatting concerns.

**Quality:**

3

**Strengths And Weaknesses:**

(+) = strength, (-) = weakness

**Quality**:
1. (+) The solution is technically sound, and the design choices are well validated through ablation studies.
2. (+) The results appear strong compared to existing methods and cover multiple datasets.
3. (+) The approach remains very simple by using an existing mechanism in transformer models, yet is very powerful.
4. (-) I find the way some motivations, properties, and results are presented to be potentially misleading:
- 4.1. While I agree in the value of this work, I struggle to understand how it resolves the claimed issues about standard two-stage approaches. The authors claim that two-stage approaches may not achieve global optimality, and that specific approaches such as NetVLAD might suffer from cluster centres learned on the training set not being suitable for the test set. However, two-stage approaches can also be fully differentiable, and I do not see why ImAge would be more likely to achieve global optimality. Moreover, I also do not see any reason or results indicating that ImAge would not also suffer from differences between train and test settings.
- 4.2. The claims on the approach being aggregator-free are a bit misleading. While I agree this could be considered an implicit aggregation mechanism, as it is already part of the original model, I still believe that calling the method "aggregator-free" is not completely correct and confusing. Similarly, it doesn't really have 0 aggregator parameters, as the shared attention weights are fine-tuned to learn the aggregation.
- 4.3. While the results in Table 2 are interesting, they are not based on a fair comparison, as mentioned in the text. I believe this should be made clearer by specifying it in the caption as well.
- 4.4. The results of figures 4, 5, and 6 do not seem very representative of overall performance. The baseline results on the datasets presented were still strong, and usually just slightly below the performance of ImAge, and it is not specified how these examples were selected. As such, I believe these figures might only be highlighting cherry-picked samples where ImAge succeeded and the baselines did not, but I do not believe we can make many conclusions based on this. If the samples are actually randomly chosen, it should be clarified. Based on the overall metrics, there are likely cases where ImAge also fails and some baselines succeed. I would expect a more complete quantitative study on the performance with different types of scenarios (e.g., dynamic objects, viewpoint variations, etc.) if the authors want to claim that ImAge consistently retrieves the correct results where others don't (Fig. 4 caption).
- 4.5. In Table 2, CricaVPR and SuperVLAD are not included in the bold/underlined results only on Pitts30k, but it would be more consistent to either not include them on any datasets, or include them on all.

**Clarity**:
1. (+) The introduction of the idea of the paper relative to existing works is very clear, and the figures illustrating the approach are easy to understand.
2. (-) The text in the paper is very dense, with figure captions sometimes getting really close to the text (e.g., Fig. 3). This negatively affects how easy it is to read the paper, and breaking down long paragraphs with more structure would help improve clarity.
3. (-) The details of the k-means clustering used for initialising tokens remain unclear to me (see question 2).

**Significance**:
1. (+) I believe that the results from this paper can affect what future research on the topic of visual place recognition will focus on, potentially shifting away from designing new aggregation mechanisms and focusing on best adapting existing pre-trained models and relying on their existing aggregation mechanisms.

**Originality**:
1. (+) While the technical solution proposed relies on existing mechanisms, the strength of the results combined with the simplicity of the approach makes this work original enough in the context of the visual place recognition literature.

---

> ### Author Rebuttal · Authors · 2025-07-30
>
> Thanks for your positive recommendation and insightful comments. The following are Responses to the Questions.
>
> **Q1 The comment on quality #4**
>
> -	For quality 4.1: Sorry to confuse you. **a)** The claim of “global optimality” is indeed inaccurate because other methods like NetVLAD can also be fully differentiable. We will delete this claim and directly state that “our method provides a novel perspective different from the previous paradigm, unifying feature extraction and aggregation into a more cohesive framework”. Unlike previous methods where the backbone is only used for patch feature extraction, our global features based on implicit aggregation can be further corrected and refined synchronously with the patch token extraction process in the backbone, achieving better performance effectively and efficiently. **b)** Regarding “NetVLAD might suffer from inappropriate cluster centres”, we mainly want to illustrate that it is difficult to artificially design a perfect aggregator, which motivates us to propose a  solution that does not require the explicit design of additional aggregators. Since domain generalization is not the main focus of our work, we will remove this statement to avoid causing confusion.
>
> -	For quality 4.2: We agree with you. "aggregator-free" and “0 aggregator parameters” are a bit confusing. We will provide further explanations in these places -- there is no **extra explicit** aggregator, and our aggregation process is completed in the original backbone.
>
> -	For quality 4.3: We will add clearer explanations in the table caption according to your suggestion.
>
> -	For quality 4.4: You are right. These examples are cherry-picked. In fact, this is a common practice in many great VPR works, e.g., in the main paper or supplementary material of the CosPlace, EigenPlaces, StructVPR, and CricaVPR works. To demonstrate the performance of methods as reasonably as possible, we have included a wide range of scenarios (viewpoint and/or condition changes, dynamic objects, etc.) in Figures 4 and 5. Additionally, Figure 6 presents the failure cases of our approach. Following your suggestion, we will add some cases where ImAge fails while some baselines succeed and provide further discussion.
>
> -	For quality 4.5: We will revise it according to your suggestion.
>
> **Q2. How clustering works**
>
> The clustering process is basically the same as the k-means clustering in the NetVLAD implementation of the VG Benchmark work [1]. We randomly sample some training images (e.g., 500 images), and use the pre-trained backbone to extract patch tokens (features) from these images. Then we randomly sample some patch tokens of each image (e.g., 100 tokens per image). Finally, we perform k-means clustering for the obtained (500*100) patch tokens. The transferring to the new settings is very easy. This is demonstrated by the widely used VG benchmark work (ours is the same).
>
> **Q3. Potential limitations**
>
> Thanks for your insightful comments. The proposed method is indeed not a good choice when we want to keep the backbone frozen or when the backbone is extremely expensive to fine-tune. We will discuss these two points in the final paper. However, it is worth noting that we only fine-tune the last few blocks of the transformer backbone, which is relatively cheap. This can be seen from the response to the Q1 of `Reviewer VZG8`. We can also further reduce the fine-tuning cost by reducing the number of blocks fine-tuned, but this may cause performance degradation.
>
> **Q4. Why do tokens represent separate categories?**
>
> Sorry to confuse you. We guess the word "object category" misleads you. We originally intended to use the analogy of human-level semantic classification to visually illustrate the division of multiple agg tokens, which is indeed not rigorous enough. The supplementary materials we submitted (a PDF in the zip file) have shown the attention visualization of agg tokens to image patches, which provides vivid and interesting qualitative results. The highlight areas of attention for different agg tokens are indeed different, that is, each token is responsible for representing different categories. In fact, both the initial k-means-based initialization and subsequent training will make different tokens represent different categories. If they represent the same category, multiple tokens will be redundant. Only when different tokens represent different categories can the representation ability be stronger. However, it does not mean that a token exactly corresponds to a human semantic object (e.g., the building or vegetation). We can see that some tokens contain both buildings and vegetation, and there are also multiple tokens that focus on buildings. To avoid misleading, we think a reasonable expression could be "each added agg token representing a unique category (but not exactly an object category in human semantics, such as building or vegetation)".
>
> **Q5. Are aggregation tokens "output-oriented tokens"**
>
> Your judgment is correct—the proposed aggregation tokens are indeed "output-oriented tokens." In fact, we explicitly mentioned this point in the initial manuscript, but it was deleted during the paper’s streamlining process. We only stated that our work was inspired by the DINOv2-register work, not that our method belongs to that category. Since this has caused ambiguity, we will clearly state in the final paper that our method falls into the "output-oriented tokens" category.
>
> **Clarity. The paper is very dense**
>
> We will make revisions according to your suggestions.
>
> **Thanks again for your valuable comments and suggestions. Please let us know if you have any further concerns. We'd be very glad to address them.**
>
> **`Reference`**
>
> [1] Gabriele Berton, et al. Deep visual geo-localization benchmark. CVPR 2022.

---

> > ### Comment · Reviewer_WgNP · 2025-08-04
> >
> > I thank the authors for their careful consideration of my comments.
> >
> > I believe that they have addressed all my concerns, and that their proposed updates will improve the paper.

---

> > > ### Author Response · Authors · 2025-08-05
> > >
> > > We are glad that our response has addressed all your concerns. Thanks a lot for your comments and suggestions to improve our paper. We will incorporate them into the final version.

---

### Note · Authors · 2025-08-12

Dear AC and Reviewers,

Thanks a lot for your valuable time and effort to review our work. This final remark is to summarize the Rebuttal/Discussion and convey some information to Reviewer zpn8.

Three reviewers gave our paper a positive recommendation in the initial review (initial rating: 5, 5, 4). They appreciated our work in terms of novelty, SOTA performance, thorough ablation studies, convincing visualization results (of the attention weights of agg tokens), etc. The comments made by Reviewer WgNP and Reviewer tSze mainly concern some inappropriate claims in our paper. We have addressed these issues in Rebuttal and received confirmation. For the comments from Reviewer VZG8, we have also provided detailed responses and supplementary results, which further show the contribution of our work.

Reviewer zpn8 mainly expressed concerns about the novelty (the difference from prompt tuning) in the initial review. In Rebuttal, we clarified the differences between these two approaches in terms of purpose, methodology, implementation, and results, demonstrating the novelty of our work. Although there are still some concerns in the further feedback of Reviewer zpn8, such as the lack of visualization results to explain the role and mechanism of agg tokens. We provide a further detailed response to address these concerns in our feedback, for example, the Supplementary Material (not Appendix) we submitted contains exactly the visualization results that the reviewer expected. We would like to believe that these detailed responses can address the concerns. Since the reviewer-author discussion period has ended (Due: Aug 8 AoE), we are unable to view any new comments or provide additional responses. If you still have any concerns, we sincerely hope you could refer to our replies to other reviewers, as these replies might also address your concerns.

Thanks again to AC and all reviewers.

Best Regards,

Authors

---

### Decision · Program_Chairs · 2025-09-17

**Decision:**

Accept (poster)

**Comment:**

This paper introduces ImAge, a new framework for Visual Place Recognition (VPR). Instead of adding a separate pooling or aggregation step on top of a Vision Transformer backbone, ImAge uses special "aggregation tokens". These tokens are inserted at a chosen layer within the transformer. They use the existing self-attention mechanism to combine information from the image. The output embeddings from these tokens directly serve as the global image descriptors for recognizing places.

The main contribution is an innovative implicit aggregation mechanism. This approach eliminates the requirement for explicit, standalone aggregators (like NetVLAD or GeM). By leveraging the transformer's native self-attention through aggregation tokens, it avoids critical issues found in traditional "backbone + aggregator" systems. These issues include sub-optimal separate optimization and potential loss of spatial information during aggregation. The proposed ImAge framework demonstrates superior performance over previous VPR methods.

The paper received one “Reject” rating, and three “Accept” ratings.

Reviewer WgNP thought the authors' rebuttal had addressed the issues with presentation and clarity, the authors validated their specific design and demonstrated strong results influential for future VPR research. Reviewer WgNP kept the "Accept".

Reviewer VZG8 thought the comments from the authors resolved the concerns about W1 and W2. Reviewer VZG8 finally gave the "Accept".

Reviewer zpn8, who gave a "Reject" rating, thought the authors' rebuttals did not impress him/her substantively. Reviewer zpn8 insisted that many papers had proposed similar CLS tokens and insertion strategies in Transformers. Reviewer zpn8 questioned the fundamental novelty of ImAge's aggregation tokens, viewing them as derivative engineering adjustments akin to prompt tuning (VPT).

Reviewer tSze thought the rebuttal had addressed all the concerns, mainly the confusion in some of the writing and to remove some unjustified claims. Reviewer tSze remained the "Accept".

Reviewers WgNP, VZG8, tSze all gave the “Accept” ratings. Although Reviewer zpn8 thought the novelty is not sufficient, the experiments show impressed results and the code will be publicly available. Therefore, the paper can be accepted.